# Control of NAD⁺ homeostasis by autophagic flux modulates mitochondrial and cardiac function

Quanjiang Zhang[1,2,9], Zhonggang Li[2,3,9], Qiuxia Li[1,2], Samuel AJ Trammell [4,5], Mark S Schmidt [5],
Karla Maria Pires [6], Jinjin Cai[6], Yuan Zhang[1,2], Helena Kenny[2], Sihem Boudina [6,7],
Charles Brenner [5,8] & E Dale Abel [1,2,5 ✉]

## Abstract

Impaired autophagy is known to cause mitochondrial dysfunction and heart failure, in part due to altered mitophagy and protein quality control. However, whether additional mechanisms are involved in the development of mitochondrial dysfunction and heart failure in the setting of deficient autophagic flux remains poorly explored. Here, we show that impaired autophagic flux reduces nicotinamide adenine dinucleotide (NAD⁺) availability in cardiomyocytes. NAD⁺ deficiency upon autophagic impairment is attributable to the induction of nicotinamide N-methyltransferase (NNMT), which methylates the NAD⁺ precursor nicotinamide (NAM) to generate N-methyl-nicotinamide (MeNAM). The administration of nicotinamide mononucleotide (NMN) or inhibition of NNMT activity in autophagy-deficient hearts and cardiomyocytes restores NAD⁺ levels and ameliorates cardiac and mitochondrial dysfunction. Mechanistically, autophagic inhibition causes the accumulation of SQSTM1, which activates NF-κB signaling and promotes *NNMT* transcription. In summary, we describe a novel mechanism illustrating how autophagic flux maintains mitochondrial and cardiac function by mediating SQSTM1-NF-κB-NNMT signaling and controlling the cellular levels of NAD⁺.

**Keywords** Autophagic Flux; NAD⁺ Metabolism; Mitochondrial Homeostasis; Heart Dysfunction
**Subject Categories** Autophagy & Cell Death; Cardiovascular System; Metabolism

## Introduction

Autophagy is a ubiquitous catabolic process by which cells engulf and deliver damaged proteins and organelles to lysosomes for degradation, maintaining cellular homeostasis (Rubinsztein et al, 2011; Yamaguchi, 2019). Autophagy has been extensively implicated in regulating cardiac structure and function under various physiological and pathophysiological conditions, such as ischemic heart disease, heart failure, and heart hypertrophy (Bhuiyan et al, 2013; Kassiotis et al, 2009; Lavandero et al, 2015; Nakai et al, 2007; Riehle et al, 2013; Sciarretta et al, 2012; Xie et al, 2014), and altered autophagy has been involved in the development of some inherited cardiomyopathies, such as Danon's disease in which autophagy is impaired secondary to the deficiency of lysosome-associated membrane protein 2 (LAMP-2) (Nascimbeni et al, 2017), left ventricular noncompaction (LVNC) secondary to PLEKHM2 mutations (Muhammad et al, 2015), and desmin-related cardiomyopathy (DRM) secondary to desmin mutations (Bhuiyan et al, 2013). In addition, impaired autophagy has been demonstrated to directly induce cardiomyopathy (Nakai et al, 2007; Taneike et al, 2010). As such, the mechanisms underlying the alteration of autophagic flux have been extensively investigated in hearts under various physiological and pathophysiological conditions (Sciarretta et al, 2015; Zhang et al, 2017). Although impaired mitophagy and protein quality control may contribute to mitochondrial and cardiac dysfunction, a gap in knowledge remains concerning mechanisms linking impaired autophagic flux and the development of cardiac dysfunction.

An important proposed mechanism underlying cardiac dysfunction in the setting of autophagic defects is the accumulation of damaged cytosolic materials, including organelles, that directly induce cellular cytotoxicity leading ultimately to cell death. Accumulation of damaged proteins (misfolded, polyubiquitinated, oxidized, or aggregated) impairs their function and those of other proteins to alter homeostatic signaling, metabolites, and ion fluxes (Bhuiyan et al, 2013; Nakai et al, 2007; Taneike et al, 2010). Dysfunctional mitochondria that accumulate in the face of reduced mitophagy induce ROS production and release of cytochrome *c* to initiate apoptotic cell death (Bhuiyan et al, 2013; Dhingra et al, 2019; Nakai et al, 2007; Taneike et al, 2010; Wu et al, 2009). Increased release of mtDNA may activate inflammatory responses and cell death pathways in pressure-overloaded hearts,

[1]Division of Endocrinology, Diabetes and Metabolism, Department of Medicine, David Geffen School of Medicine and UCLA Health, University of California-Los Angeles, Los Angeles, CA 90095, USA. [2]Department of Internal Medicine, Fraternal Order of Eagles Diabetes Research Center, and Abboud Cardiovascular Research Center, Carver College of Medicine, University of Iowa, Iowa City, IA 52242, USA. [3]Department of Human Genetics, School of Medicine, University of Utah, Salt Lake City, UT 84112, USA. [4]Department of Biomedical Sciences, University of Copenhagen, 2200 Copenhagen, Denmark. [5]Department of Biochemistry, Carver College of Medicine, University of Iowa, Iowa City, IA 52242, USA. [6]Division of Endocrinology, Metabolism and Diabetes, and Program in Molecular Medicine, School of Medicine, University of Utah, Salt Lake City, UT 84112, USA. [7]Department of Nutrition and Integrative Physiology, College of Health, University of Utah, Salt Lake City, UT 84112, USA. [8]Department of Diabetes & Cancer Metabolism, City of Hope National Medical Center, Duarte, CA 91010, USA. [9]These authors contributed equally: Quanjiang Zhang, Zhonggang Li. ✉E-mail: DOMChair_DaleAbel@mednet.ucla.edu

contributing to the development of heart failure (Oka et al, 2012; Wu et al, 2009).

In addition to removing damaged cytosolic materials, autophagy also selectively degrades specific protein(s) involved in metabolism or signal transduction by binding to the SQSTM1-motif that targets these peptides to the autophagic machinery. Pleiotropic functions of SQSTM1, also known as the ubiquitin-binding protein p62, have been described, including binding to MEKK3 and Raptor to regulate MTORC1 signaling, TRAF6 to regulate NF-κB signaling, MAP1LC3A (LC3) to regulate autophagic loading, and KEAP1 to regulate NRF2 signaling (Moscat et al, 2016). Impaired autophagy has been linked to altered function of these signaling pathways. In addition, impaired autophagy has been shown to deplete NAD$^+$ levels and thus cause cell death (Kataura et al, 2022; Wilson et al, 2023). This therefore leaves open the possibility that in addition to accumulation of damaged cytosolic materials, the disruption of the above-mentioned mechanisms could contribute to mitochondrial and cardiac dysfunction in the setting of autophagic deficiency. These additional mechanisms have not to date been rigorously evaluated in the context of heart failure.

In this study, we found that in the setting of impaired autophagy induced by ATG3 deficiency, hearts developed NAD$^+$ deficiency attributed to accelerated NAD$^+$ clearance, but not to decreased NAD$^+$ salvage, de novo synthesis, or increased NAD$^+$ consumption, which contributes to the development of mitochondrial and cardiac dysfunction and resultant heart failure. The acceleration of NAD$^+$ clearance is due to induction of nicotinamide N-methyltransferase (NNMT) attributable to the activation of SQSTM1-NF-κB signal transduction. These data indicate that autophagy maintains mitochondrial and cardiac function in part by mediating SQSTM1-NF-κB signal transduction to control cellular levels of NAD$^+$, a central gauge of nutrient status.

# Results

## Impairment of autophagic initiation specifically in cardiomyocytes causes cardiac dysfunction and early mortality

To investigate the mechanisms by which impaired autophagic flux causes heart failure, we disrupted autophagic initiation specifically in cardiomyocytes via either inducible (MYH6-MerCreMer driven) or constitutive (MYH6-Cre driven) knockout of ATG3, a key mediator of autophagic initiation that mediates the conversion of MAP1LC3A-I (LC3-I) to MAP1LC3A-II (LC3-II) (Nath et al, 2014; Yamada et al, 2007). In cardiac-specific inducible ATG3 knockout (ciAtg3 KO) mice, 1 week after the final tamoxifen administration, ATG3 protein was almost undetectable, and autophagic flux was blocked in ciAtg3 KO mouse hearts as shown by decreased LC3-II protein and accumulation of LC3-I and SQSTM1 (Fig. 1A). Mice exhibited moderate heart hypertrophy (Fig. 1B). Cardiac function was determined in lightly-sedated mice (Midazolam), which maintained heart rates at greater than 600 bpm. Under these conditions, cardiac dysfunction manifested in ciAtg3 KO mice as early as 14 weeks after the last tamoxifen injection, and mortality was increased, with all mice dying by 48 weeks of age (Fig. 1C,D). In cardiac-specific constitutive ATG3 knockout (cAtg3-KO) mice, ATG3 protein was completely deleted in isolated cardiomyocytes

(Fig. 1E), but not in other tissues such as liver, skeletal muscle, white adipose tissues, or kidney (Appendix Fig. S1A), validating cardiac-specific ATG3 knockout. The cAtg3-KO mice were born approximately at the Mendelian ratio (Appendix Table S1), implying that ATG3 loss in cardiomyocytes during embryogenesis did not induce embryonic lethality. The cAtg3-KO mouse hearts exhibited impaired autophagic flux as evidenced by decreased LC3-II protein and SQSTM1 accumulation under fed conditions, and food deprivation could not initiate autophagic flux in cAtg3-KO mouse hearts (Fig. 1F). Similar to ciAtg3 KO mice, cAtg3-KO mice exhibited moderate heart hypertrophy at 4 months of age (Fig. 1G). Moreover, the mRNA expression levels of atrial natriuretic peptide (NPPA) and brain natriuretic peptide (NPPB) were both significantly increased in cAtg3-KO hearts, while phospholamban (PLN) expression was decreased (Fig. 1H), consistent with cardiac remodeling (Gruson et al, 2011; Houweling et al, 2005; Nieminen et al, 2005).

Echocardiography under conditions of light sedation revealed that beginning at 46 weeks of age, cAtg3-KO mice manifested evidence of cardiac dysfunction, relative to their WT littermates, with increasing prevalence with age (Fig. 1I). CAtg3-KO mice exhibited increased mortality with death beginning at 45 weeks of age and all animals died by 82 weeks (Fig. 1J). Impaired autophagic flux in cAtg3-KO mouse hearts did not influence MTOR or AMPKα signaling pathways (Appendix Fig. S1B), and cAtg3-KO mouse hearts did not display any signs of fibrosis and apoptosis at 16 weeks of age (Figs. EV1A and EV2A), when cardiac function was relatively preserved. However, at 40 weeks of age when ventricular dysfunction was evident, cAtg3-KO mouse hearts displayed evident fibrosis and DNA damage (Figs. EV1B and EV2B). These data indicate that impaired autophagic flux resulting from Atg3 deficiency causes cardiac dysfunction and mortality, similar to previous studies performed on Atg5-deficient mouse hearts (Nakai et al, 2007; Taneike et al, 2010), but with a distinctive feature of relatively slow development of heart failure.

## Reduced mitochondrial metabolism and biogenesis in cAtg3-KO hearts

Given the importance of autophagy in maintaining mitochondrial homeostasis, we further investigated mitochondrial function in cAtg3-KO mouse hearts. At 16 weeks of age, cAtg3-KO mouse hearts exhibited mitochondrial dysfunction as evidenced by reduced activities of citrate synthase (CS), a general index of mitochondrial oxidative capacity (Larsen et al, 2012) (Fig. 2A); impaired palmitoyl-carnitine supported ADP-stimulated oxygen consumption ($V_{ADP}$) (Fig. 2B); and decreased ATP production (Fig. 2C). ATP/O ratios were unchanged (Fig. 2D), indicating preserved mitochondrial coupling. In addition, steady-state levels of the citric acid cycle intermediates, acetyl-CoA and succinyl-CoA were markedly decreased (Fig. 2E). Studies have demonstrated that impaired autophagy, mitochondrial dysfunction, and abnormal fatty acid metabolism commonly coexist (Dong and Czaja, 2011; Gumucio et al, 2019; Schulze et al, 2016; Zhang et al, 2018). CAtg3-KO mouse hearts exhibited decreased expression levels of fatty acid oxidation (FAO) genes (Fig. EV3A), decreased carnitine palmitoyl-transferase (CPT) activity (Fig. 2F), increased myocardial triglycerides (TGs) accumulation (Fig. 2G), and increased neutral lipids (Fig. EV3B), consistent with decreased fatty acid oxidation. These

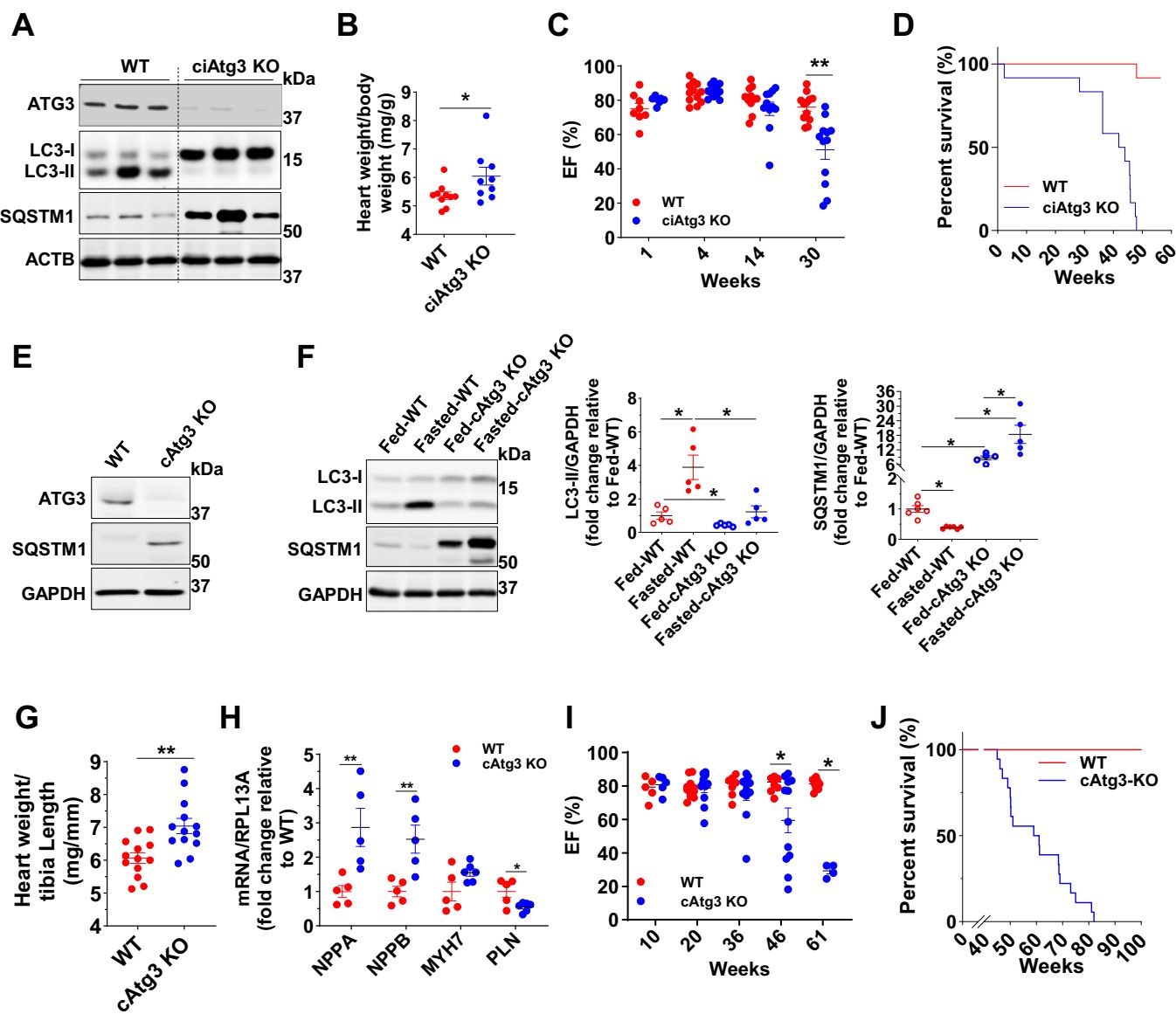

**Figure 1. Cardiac-specific *ATG3* deletion blocks autophagy and precipitates cardiac contractile dysfunction**

(**A**) ATG3, LC3 (MAP1LC3A), SQSTM1, and beta-actin (ACTB) protein levels in WT and ciAtg3 KO mouse hearts 1 week after last tamoxifen injection. Mice, at 6 weeks of age, were intraperitoneally injected with tamoxifen at a dose of 100 μg/g body weight/day for 5 consecutive days. One week after last tamoxifen injection, mice were euthanized and hearts were harvested for experiments. Age-matched *ATG3^{loxP/loxP}* mice without Cre were injected with the same amount of tamoxifen as WT control. Representative blots are shown. $n = 3$ per group. (**B**) Heart weight of WT and ciAtg3 KO mice, 1 week after last tamoxifen injection. $n = 7$–8. Data are mean ± SEM. An unpaired *t* test was used to determine statistical significance between two groups. *$P < 0.05$. (**C**) Ejection fraction in WT and ciAtg3 KO mice 1, 4, 14, and 30 weeks after last tamoxifen injection. Measurements were performed under light sedation with midazolam. $n = 7$–12 per group. Data are mean ± SEM. Unpaired *t* tests were used to determine statistical significance between two groups at corresponding time points. **$P < 0.01$. (**D**) Survival curve of WT and ciAtg3 KO mice after the last tamoxifen injection. $n = 12$ per group. (**E**) ATG3, SQSTM1, and GAPDH protein levels in cardiomyocytes. Cardiomyocytes were enzymatically isolated from WT and cAtg3-KO mouse hearts. Representative blots are shown. This experiment was repeated twice independently. (**F**) LC3, SQSTM1, and GAPDH protein levels in mouse hearts. Mice were either randomly fed or fasted for 48 h, $n = 5$ per group. Data are mean ± SEM. One-way ANOVA followed by Bonferroni's multiple comparison tests was used to determine statistical significance. *$P < 0.05$. (**G**) Heart weight of WT and cAtg3-KO mice at 16 weeks of age. $n = 13$ per group. Data are mean ± SEM. An unpaired *t* test was used to determine statistical significance between two groups. **$P < 0.01$. (**H**) *NPPA*, *NPPB*, *MYH7*, and *PLN* mRNA expression levels in WT and *ATG3* KO mouse hearts at 16 weeks of age, $n = 5$ per group. Unpaired *t* tests were used to determine statistical significance between two groups. Data are mean ± SEM. **$P < 0.01$. (**I**) Ejection fraction in WT and cAtg3-KO mice at ages as indicated. Measurements were performed under light sedation with midazolam. $n = 4$–12 per group. Unpaired *t* tests were used to determine statistical significance between two groups at corresponding time points. Data are mean ± SEM. *$P < 0.05$. (**J**) Survival curve of WT and cAtg3-KO mice. $n = 18$. Data information: Cardiomyocytes for (**E**) were enzymatically isolated from 16-week-old WT and cAtg3-KO mice at a time when cardiac function was preserved in cAtg3-KO (**I**). After being enzymatically isolated, cardiomyocytes were immediately lysed in ice-cold homogenization buffer without any pre-treatments or culture. This experiment was repeated twice independently. Source data are available online for this figure.

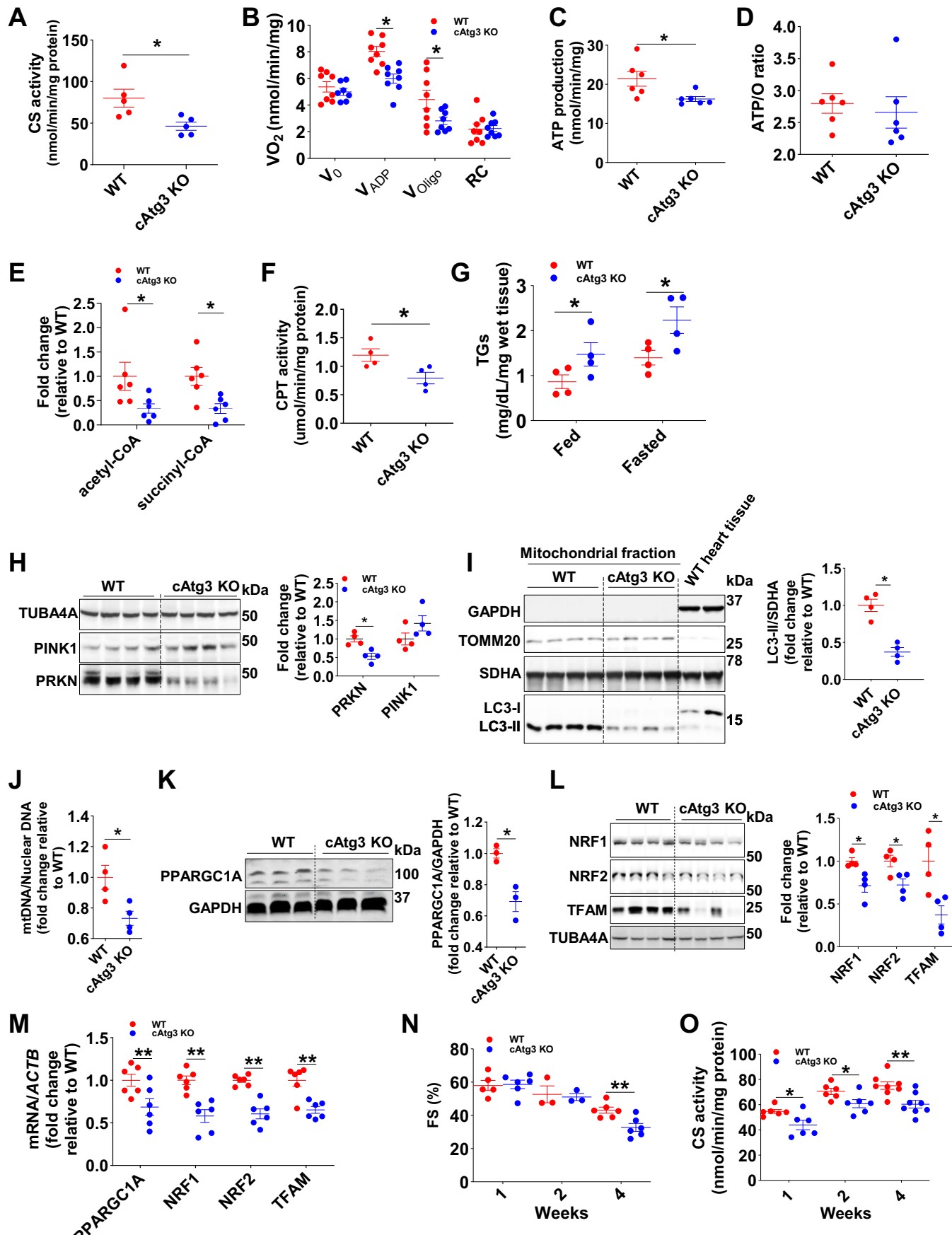

**Figure 2. CAtg3-KO mouse hearts exhibit mitochondrial dysfunction and myocardial lipid accumulation.**

(A) Citrate synthase (CS) activity was measured in 16-week-old WT and cAtg3-KO mouse heart lysates. $n = 5$ per group. An unpaired $t$ test was used to determine statistical significance between two groups. Data are mean ± SEM. *$P < 0.05$. (B) Oxygen consumption rate was measured in saponin-permeabilized cardiac fibers from 16-week-old WT and cAtg3-KO mouse hearts. $n = 8$ per group. Unpaired $t$ tests were used to determine statistical significance between two groups at corresponding substrates. Data are mean ± SEM. *$P < 0.05$. (C) ATP production rate was measured in cardiac fibers following stimulation with ADP. $n = 6$ per group. Data are mean ± SEM. An unpaired $t$ test was used to determine statistical significance between two groups. *$P < 0.05$. (D) ATP/O ratio measured in saponin-permeabilized cardiac fibers from 16-week-old WT and cAtg3-KO mice. $n = 6$ per group. Data are mean ± SEM. (E) Acetyl-CoA and Succinyl-CoA levels in 4-week-old WT and Atg3 KO mouse hearts. $n = 5–6$ per group. Data are mean ± SEM. An unpaired $t$ test was used to determine statistical significance between two groups. *$P < 0.05$. (F) Carnitine palmitoyltransferase (CPT) activity in mitochondria. Mitochondria were isolated from 16-week-old WT and cAtg3-KO mouse hearts. $n = 4$ per group. An unpaired $t$ test was used to determine statistical significance between two groups. Data are mean ± SEM. *$P < 0.05$. (G) Triglycerides (TGs) levels in 16-week-old WT and cAtg3-KO mouse hearts. Mice were randomly fed or subjected to a 48 h. food deprivation (fasted). $n = 4$ per group. Unpaired $t$ tests were used to determine statistical significance between two groups. Data are mean ± SEM. *$P < 0.05$. (H) TUBA4A (alpha-tubulin), PINK1, and PRKN (Parkin) protein levels in WT and cAtg3 KO mouse hearts at 16 weeks of age. $n = 4$ per group. Unpaired $t$ tests were used to determine statistical significance between two groups. Data are mean ± SEM. *$P < 0.05$. (I) GAPDH, TOMM20, SDHA, and LC3 (MAP1LC3A) protein levels in mitochondrial fractions of WT and cAtg3-KO mouse hearts at 16 weeks of age. $n = 4$ per group. An unpaired $t$ test was used to determine statistical significance between two groups. Data are mean ± SEM. *$P < 0.05$. (J) MtDNA copy number in 16-week-old WT and cAtg3-KO mouse hearts. $n = 4$ per group. An unpaired $t$ test was used to determine statistical significance between two groups. Data are mean ± SEM. *$P < 0.05$. (K) PPARGC1A (PGC1 alpha) protein levels in WT and cAtg3-KO mouse hearts at 16 weeks of age, $n = 3$ per group. An unpaired $t$ test was used to determine statistical significance between two groups. Data are mean ± SEM. *$P < 0.05$. (L) Protein levels of NRF1, NRF2, and TFAM in WT and cAtg3-KO mouse hearts at 16 weeks of age, $n = 4$ per group. Unpaired $t$ tests were used to determine statistical significance between two groups. Data are mean ± SEM. *$P < 0.05$. (M) MRNA expression levels of mitochondrial biogenesis genes in WT and cAtg3-KO mouse hearts at 16 weeks of age, $n = 6$ per group. Unpaired $t$ tests were used to determine statistical significance between two groups. Data are mean ± SEM. **$P < 0.01$. (N) Time course of fractional shortening measured in isoflurane-anesthetized WT and cAtg3-KO mice at 1, 2, and 4 weeks of age, $n = 6$ per group. Unpaired $t$ tests were used to determine statistical significance between two groups at corresponding time points. Data are mean ± SEM. **$P < 0.01$. (O) Time course of citrate synthase (CS) activity in WT and cAtg3-KO mouse hearts at 1, 2, and 4 weeks of age, $n = 6–8$ per group. Data are mean ± SEM. Unpaired $t$ tests were used to determine statistical significance between two groups at corresponding time points. *$P < 0.05$, **$P < 0.01$. Data information: (B–D) ATP levels and oxygen consumption were measured in saponin-permeabilized cardiac fibers after the addition of 1 mM ADP. (I) LC3-I is almost undetectable. Last two lanes are whole heart homogenates as controls for mitochondrial fractions. Source data are available online for this figure.

data indicate that autophagy plays an essential role in maintaining mitochondrial function and subsequent fatty acid metabolism.

To investigate the mechanisms underlying mitochondrial dysfunction in cAtg3-KO hearts, we determined parameters of mitochondrial turnover (mitophagy) and mitochondrial biogenesis. At 16 weeks of age, Parkin (PRKN) protein was significantly repressed in cAtg3-KO mouse heart tissue, and LC3-II protein expression was decreased in mitochondrial fractions from cAtg3-KO mouse hearts (Fig. 2H,I). Furthermore, cAtg3-KO mouse hearts exhibited reduced mitochondrial DNA copy number (Fig. 2J), decreased protein expression of PGC1α (PPARGC1A), NRF1, NRF2, and TFAM; and reduced gene expression of *PPARGC1A*, *NRF1*, *NRF2*, and *TFAM* (Fig. 2K–M). These data reveal decreased mitophagy and mitochondrial biogenesis in cAtg3-KO mouse hearts. Thus, reduced mitophagy and impaired mitochondrial biogenesis could concurrently contribute to mitochondrial dysfunction in cAtg3-KO mouse hearts.

## Mitochondrial dysfunction precedes age-dependent cardiac dysfunction in cAtg3-KO mouse hearts

Mitochondrial dysfunction has been well-known to coexist with heart failure, and our prior studies indicated that in certain models, mitochondrial dysfunction may precede cardiac dysfunction (Riehle et al, 2013). We, therefore, monitored cardiac function in a cohort of cAtg3-KO mice and WT littermates after birth. These studies performed under isoflurane anesthesia (average heart rate 400 bpm), detected cardiac dysfunction as evidenced by decreased fractional shortening starting at 4 weeks of age in cAtg3-KO mice (Fig. 2N). However, CS activity was reduced as early as 1 week of age when cardiac function was still preserved, and this reduction in CS activity persisted at 4 and 16 weeks of age in cAtg3-KO mouse hearts (Fig. 2O,A). These data confirm that mitochondrial dysfunction precedes and coexists with cardiac dysfunction in cAtg3-KO mice.

## Glycolysis is impaired at the level of GAPDH in cAtg3-KO mouse hearts

The metabolomics profile of glycolytic intermediates was altered in cAtg3-KO mouse hearts (Fig. 3A). Above the level of glyceraldehyde 3-phosphate dehydrogenase (GAPDH), the abundance of glycolytic intermediates including glucose, glucose-6-phosphate, glucose-1-phosphate, fructose-6-phosphate, and dihydroxyacetone-phosphate was significantly increased in 1-week-old cAtg3-KO mouse hearts relative to age-matched WT mice and was further augmented in 4-week-old cAtg3-KO hearts (Fig. 3A). Downstream of GAPDH, glycolytic intermediates including 1,3-bisphosphoglycerate, 3-phosphoglycerate, 2-phosphoglycerate, phosphoenolpyruvate, pyruvate, and lactate were not correspondingly increased and, in some cases, were decreased. Thus, the overrepresentation of molecules from the preparatory stage of glycolysis raised the possibility that glycolysis could be backed up due to inhibition of GAPDH activity.

## Atg3 KO mouse hearts have depressed GAPDH activity and NAD$^+$ levels

We next determined that the forward GAPDH activity in cardiac extracts was significantly decreased in cAtg3-KO mice with respect to wild type at 4 weeks of age (Fig. 3B), despite normal levels of GAPDH protein (Fig. 3C). However, the addition of 10 mM exogenous NAD$^+$ normalized GAPDH activity in cAtg3-KO to that of the WT littermates (Fig. 3B). We then measured cytosolic NAD$^+$ content in cAtg3-KO hearts and indeed observed significantly decreased cytosolic NAD$^+$ content (Fig. 3D). To explore if NAD$^+$ depletion correlated with reduced autophagic flux, we administered chloroquine (CQ), which blocks autophagosome–lysosome fusion, to WT mice and measured cytosolic NAD$^+$ content in hearts. Similar to cAtg3-KO hearts, cytosolic NAD$^+$ content was significantly decreased in CQ-treated mouse hearts (Fig. 3E).

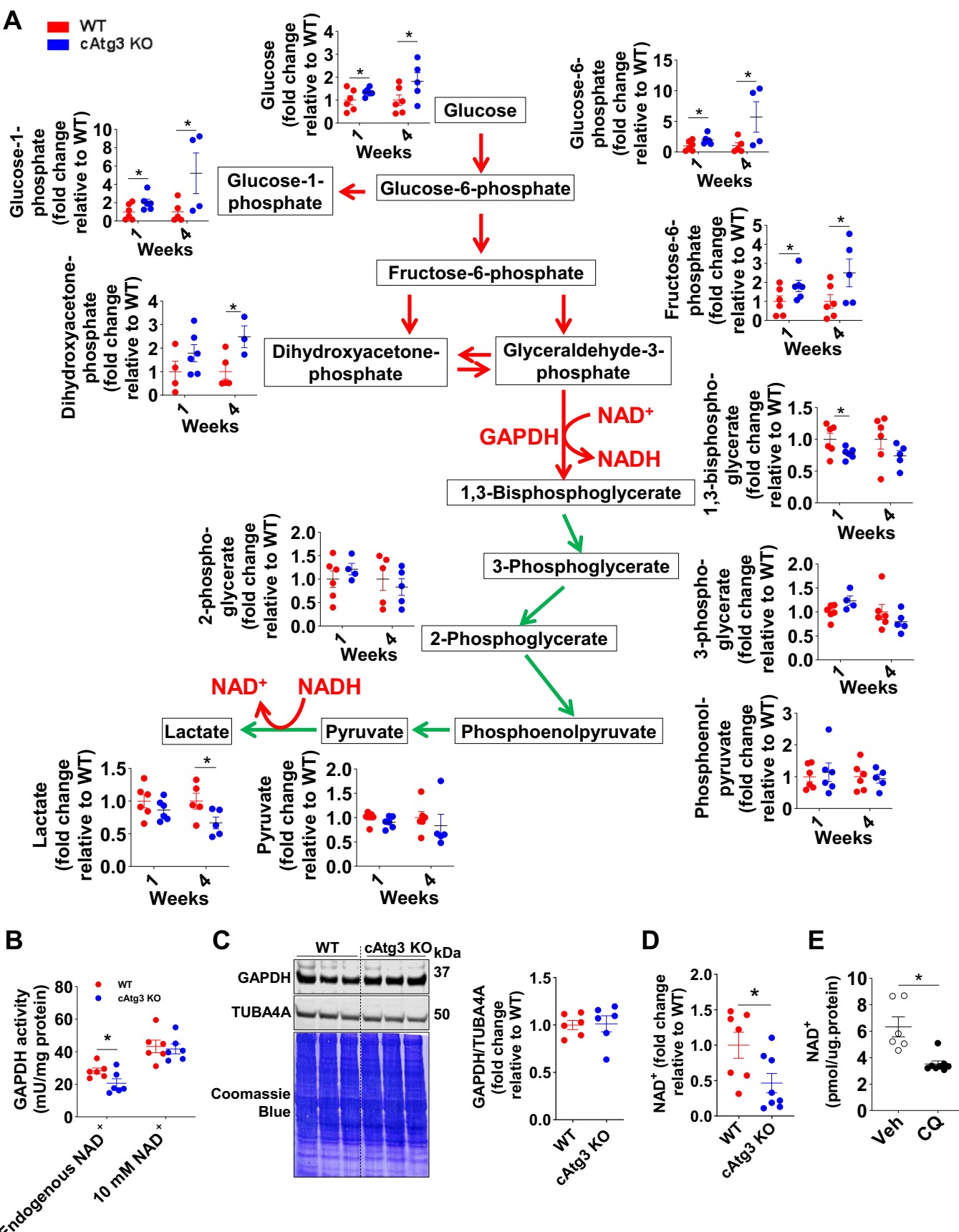

**Figure 3.   Altered glycolysis and reduced GAPDH activity in cAtg3-KO mouse hearts.**

(A) Glycolytic intermediates in 1- and 4-week-old WT and cAtg3-KO mouse hearts, shown as fold change relative to age-matched WT mice, $n = 3$ to 6 per group. Unpaired $t$ tests were used to determine statistical significance between two groups. Data are mean ± SEM. *$P < 0.05$. (B) GAPDH activity in 4-week-old WT and cAtg3-KO mouse hearts, without or with the addition of 10 mM exogenous $NAD^+$, $n = 6$ per group. Unpaired $t$ tests were used to determine statistical significance between two groups. Data are mean ± SEM. *$P < 0.05$. (C) GAPDH and TUBA4A (alpha-tubulin) protein levels in 4-week-old WT and cAtg3-KO mouse heart homogenates, $n = 6$ per group. After immunoblotting, the membrane was stained in Coomassie Blue. (D) $NAD^+$ levels in 6-week-old WT and cAtg3-KO mouse hearts. $n = 7$–8 per group. Data are mean ± SEM. An unpaired $t$ test was used to determine statistical significance between two groups. *$P < 0.05$. (E) $NAD^+$ levels in WT mouse hearts after administration of chloroquine (CQ). Mice were used at ~12 weeks of age. $n = 6$–8 per group. Data are mean ± SEM. An unpaired $t$ test was used to determine statistical significance between two groups. *$P < 0.05$. Data information: GAPDH activity (B) was measured in 4-, but not 16-, week-old mouse hearts without or with the addition of 10 mM exogenous $NAD^+$. Source data are available online for this figure.

## NAD⁺ deficiency contributes to cardiac dysfunction in cAtg3-KO mouse Hearts

To further explore if $NAD^+$ deficiency could underlie mitochondrial and cardiac dysfunction in cAtg3-KO mouse hearts, mice were administered β-nicotinamide mononucleotide (NMN), an intermediate in the $NAD^+$ biosynthesis pathway known to increase cardiac $NAD^+$ content (Karamanlidis et al, 2013; Yamamoto et al, 2014). Treatment was started at weaning (3 weeks old) when cardiac dysfunction could be detected by echocardiography under isoflurane anesthesia. Following 7 days of NMN administration via intraperitoneal injection, the autophagic defect in cAtg3-KO mouse hearts persisted as evidenced by decreased LC3-II protein and SQSTM1 accumulation (Fig. 4A), but the deficient states of $NAD^+$ and its related metabolite, NADH, $NADP^+$, and NADPH in cAtg3-KO mouse hearts were significantly reversed to the levels in WT mouse hearts (Fig. 4B). NMN administration restored CS activity (Fig. 4C) and partially normalized ATP levels in cAtg3-KO mice and increased ADP levels in WT mouse hearts (Fig. 4D). Importantly, elevated *NPPB* mRNA expression was normalized (Fig. 4E), as was fractional shortening, measured in isoflurane-anesthetized cAtg3-KO mice following NMN supplementation (Fig. 4F). These data support the hypothesis that $NAD^+$ deficiency may induce mitochondrial and cardiac dysfunction in cAtg3-KO mice.

## Accelerated NAD⁺ clearance in cAtg3-KO mouse Hearts

To elucidate the mechanisms underlying decreased $NAD^+$ content in cAtg3-KO mouse hearts, we performed metabolomics to map $NAD^+$ metabolism in cAtg3-KO hearts. As shown in Fig. 5A, along with decreases in $NAD^+$ content, the levels of NADP, NADPH, and NADH derived from $NAD^+$ decreased, which is accompanied by a decrease in nicotinamide (NAM), whereas the production of methyl-nicotinamide (MeNAM), a NAM clearance product, counterintuitively increased. The levels of NaR, NR, and NMN, precursors of $NAD^+$ production, did not change (Fig. 5A). This metabolic profile suggested that $NAD^+$ degradation is accelerated toward NAM to MeNAM. To rigorously test this possibility, we performed flux metabolomics to track $NAD^+$ degradation in cAtg3-KO hearts. Mice were injected intraperitoneally with the $NAD^+$ precursor, $^{13}C$,D double-labeled nicotinamide riboside (NR) ($^{13}C$,D double-labeled NR) (Fig. 5A,B) (Trammell et al, 2016a, 2016b). Relative to WT littermates, the $^{13}C$ enrichment of $NAD^+$ and NAM in cAtg3-KO mouse hearts was significantly lower after NR administration, while $^{13}C$ enrichment of MeNAM was significantly increased in cAtg3-KO mouse hearts (Fig. 5C–E), consistent with our metabolic mapping. These data indicate accelerated $NAD^+$ clearance toward NAM then to MeNAM.

Cellular $NAD^+$ content in cardiomyocytes relies on its biosynthesis dominantly through the amidated pathway and consumption through $NAD^+$ kinase (NADK), Sirtuins, poly-ADP ribose polymerases (PARPs), cluster of differentiation 38 (CD38), and bone marrow stromal cell antigen 1 (BST1) (Abdellatif et al, 2021). The protein expression of NAM phosphoribosyltransferase (NAMPT), NMN adenylyltransferase 1-3 (NMNAT1-3), NADK, Sirtuins 1–7, PARP1/2, and BST1 did not change, but CD38 protein expression significantly decreased (Appendix Figs. S2, S3A, S4A, and S4D). In addition, we measured the activities of SIRT1 and PARP1/2, two major cellular $NAD^+$-consuming proteins. PARP1/2 activities were comparable between WT and cAtg3-KO mouse hearts (Appendix Fig. S4B). In addition, the abundance of parylated proteins was comparable between WT and cAtg3-KO mouse hearts (Appendix Fig. S4C), further indicating that PARP1 activities were not increased in cAtg3-KO mouse hearts. However, SIRT1 activities significantly decreased in cAtg3-KO mouse hearts compared to WT mouse hearts, as evidenced by increased Ac-lysine levels, and hyperacetylation of forkhead box protein O1 (FOXO1), a well-known SIRT1 target (Canto and Auwerx, 2012) (Appendix Fig. S3B–E). The addition of exogenous $NAD^+$ restored SIRT1 enzymatic activities in cAtg3-KO mouse hearts, further supporting the presence of $NAD^+$ deficiency in cAtg3-KO mouse hearts (Appendix Fig. S3B). Of note, when $NAD^+$ levels were restored by injecting NMN to cAtg3-KO, increased acetylated FOXO1 levels were normalized to the levels similar to that in WT hearts (Appendix Fig. S3E), consistent with restoration of SIRT1 activity. Taken together, these data indicate that the acceleration of $NAD^+$ clearance is not attributable to increased $NAD^+$ consumption processes.

Our data indicate that cAtg3-KO mouse hearts exhibited decreased NAM content, increased MeNAM production, and preserved NMN content. NAM may be shunted to MeNAM in a *S*-adenosyl methionine-dependent reaction catalyzed by NNMT (Fig. 5A). Indeed, *NNMT* mRNA and NNMT protein levels were both significantly induced when autophagic flux was disrupted either by deleting the *ATG3* gene (Figs. 5F,G and EV4A,B) or by pharmacological inhibition of autophagic flux in mice injected with colchicine (Col) (Fig. EV4C,D) or chloroquine (CQ) (Fig. EV4E,F). To investigate whether increased NNMT expression could sufficiently depress steady state of $NAD^+$ levels, we overexpressed NNMT in H9c2 cells, an embryonic cardiomyocyte line, and measured the levels of $NAD^+$, NAM, and MeNAM. NNMT overexpression reduced cellular $NAD^+$ levels while increasing cellular MeNAM levels. MeNAM release into the culture medium was markedly increased (Fig. 5H–J), consistent with the in vivo metabolomics data. These observations indicate that induction of NNMT could contribute to $NAD^+$ deficiency.

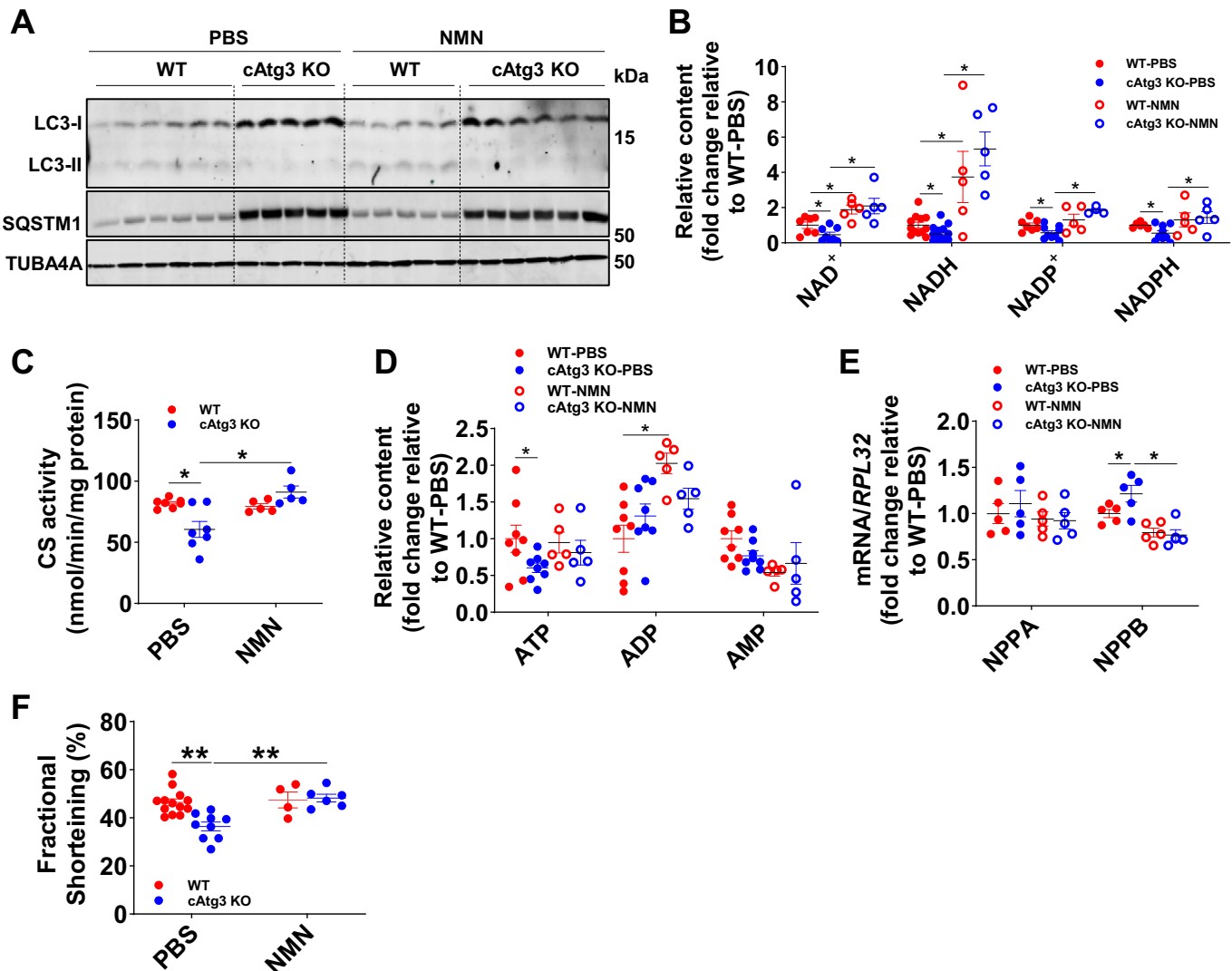

**Figure 4. Administration of β-nicotinamide mononucleotide (NMN) rescued cardiac dysfunction in cAtg3-KO mice at 4 weeks of age.**

(A) LC3 (MAP1LC3A) and SQSTM1 protein levels in PBS- or NMN-treated hearts from WT and cAtg3-KO mice, $n = 5$–6 per group. Representative blots are shown. (B) Measurements of NAD$^+$ related metabolites in hearts from WT and cAtg3-KO mice following PBS- or NMN administration, $n = 5$–7 per group. One-way ANOVA followed by Bonferroni's multiple comparison tests was used to determine statistical significance. Data are mean ± SEM. *$P < 0.05$. (C) Citrate synthase (CS) activity measurement in hearts from WT and cAtg3-KO mice following PBS- or NMN administration, $n = 5$–7 per group. One-way ANOVA followed by Bonferroni's multiple comparison tests was used to determine statistical significance. Data are mean ± SEM. *$P < 0.05$. (D) ATP, ADP, and AMP levels in PBS- or NMN-treated hearts from WT and cAtg3-KO mice. The quantitative analysis of metabolites was performed following extraction by methanol. $n = 5$–8 per group. One-way ANOVA followed by Bonferroni's multiple comparison tests was used to determine statistical significance. Data are mean ± SEM. *$P < 0.05$. (E) *NPPA* and *NPPB* mRNA expression levels in hearts from WT and cAtg3-KO mice following PBS- or NMN administration, $n = 5$ per group. One-way ANOVA followed by Bonferroni's multiple comparison tests was used to determine statistical significance. Data are mean ± SEM. *$P < 0.05$. (F) Fractional shortening determined in isoflurane-anesthetized mice following treatment with PBS or NMN $n = 4$–13 per group. Data are mean ± SEM. One-way ANOVA followed by Bonferroni's multiple comparison tests was used to determine statistical significance. **$P < 0.01$. Data information: NMN was injected intraperitoneally to 3-week-old mice at 500 mg/kg/d for 7 days. Therefore, the data in this figure were obtained in 4-week-old mouse hearts. Relative to 16-week-old cAtg3-KO mouse hearts (Fig. 1H), 4-week-old cAtg3-KO mouse hearts do not show upregulation of *NPPA*. Although induction of *NPPB* is significant, the increment is relatively lower than that in 16-week-old cAtg3-KO mouse hearts (Fig. 1H). Source data are available online for this figure.

## NAD$^+$ deficiency secondary to NNMT induction causes mitochondrial dysfunction

To determine if NAD$^+$ deficiency secondary to NNMT induction could contribute to mitochondrial dysfunction as observed in cAtg3-KO mouse hearts, we manipulated autophagic flux and measured mitochondrial respiratory function in cultured H9c2 cells. Knocking down *ATG3* or adding CQ to cells markedly

blunted autophagic flux in H9c2 cells as shown by patterns of expression of LC3-I and LC3-II and SQSTM1 accumulation (Fig. 6A,E). Studies in H9c2 cells were conducted after 16 h of CQ treatment or 48 h after siRNA transfection. These manipulations ultimately caused cell death beyond 24 h of CQ treatment or 24 h following the 48 h transfection period for siRNA-treated cells. This was evidenced by increased cleaved caspase 3 (CASP3) and decreased cell numbers (Fig. EV5). Consistent with findings in

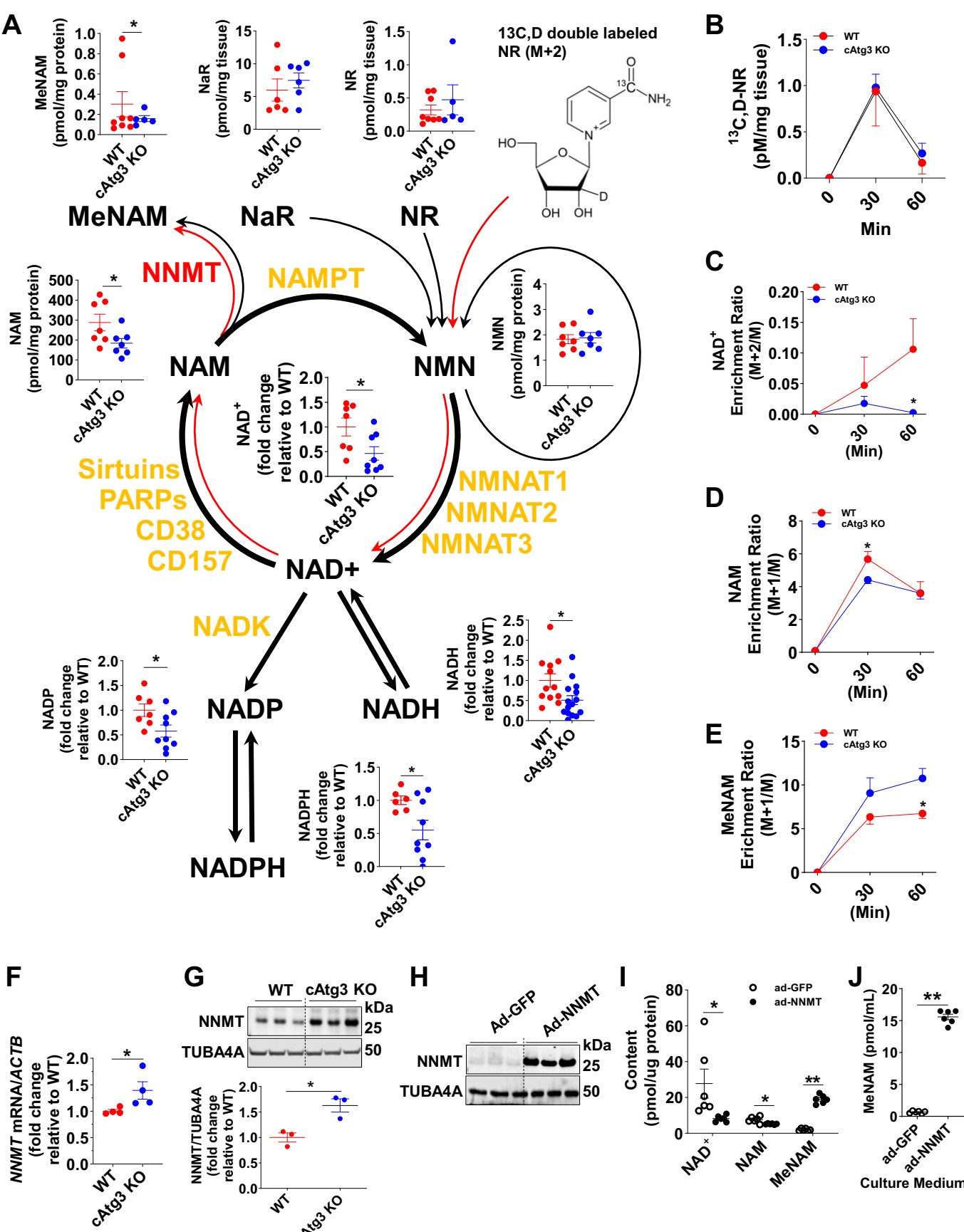

**Figure 5. CAtg3-KO mice exhibit altered NAD$^+$ metabolism in hearts.**

(A) Schematic of NAD$^+$ catabolic pathways and related metabolites, and NAD$^+$ flux analysis assessed by isotopic nicotinamide riboside (NR) in 6-week-old WT and cAtg3-KO mouse hearts. $n = 6$–13 per group. Data are mean ± SEM. Unpaired $t$ tests were used to determine statistical significance between two groups. *$P < 0.05$. (B) Double-labeled NR content in WT and cAtg3-KO mouse hearts at 0, 30 min, and 60 min after isotope-labeled NR administration, $n = 4$ per group. Data are mean ± SEM. (C–E) NAD$^+$, NAM, and MeNAM enrichment in WT and Atg3 KO hearts at 0 min, 30 min, and 60 min after isotope-labeled NR administration, $n = 4$ per group. Data are mean ± SEM. Unpaired $t$ tests were used to determine statistical significance between two groups at corresponding time points. *$P < 0.05$. (F) *NNMT* mRNA levels in WT and cAtg3-KO mouse hearts at 10 weeks of age, $n = 4$ per group. Data are mean ± SEM. An unpaired $t$ test was used to determine statistical significance between two groups. *$P < 0.05$. (G) NNMT and TUBA4A (alpha-tubulin) protein levels in WT and cAtg3-KO mouse hearts at 10 weeks of age, $n = 3$ per group. Data are mean ± SEM. An unpaired $t$ test was used to determine statistical significance between two groups. *$P < 0.05$. (H) NNMT and TUBA4A protein expression in H9c2 cells transfected with an adenovirus expressing GFP or NNMT. Representative blots are shown. $n = 3$ per group. (I) Cellular NAD$^+$, NAM, and MeNAM levels in H9c2 cells transfected with an adenovirus expressing GFP or NNMT, $n = 6$ per group. Unpaired $t$ tests were used to determine statistical significance between two groups. Data are mean ± SEM, **$P < 0.01$, *$P < 0.05$. (J) MeNAM levels in cultured medium from H9c2 cells transfected with adenovirus expressing GFP or NNMT. $n = 6$ per group. Data are mean ± SEM. An unpaired $t$ test was used to determine statistical significance between two groups. **$P < 0.01$. Data information: The data in (H–J) were obtained on H9c2 cells. This cell type is derived from embryonic BD1X rat heart tissue. Therefore, H9c2 cells are neither adult cardiomyocytes nor neonatal rat cardiomyocytes. Source data are available online for this figure.

cAtg3-KO mouse hearts, blocking autophagic flux by knocking down *ATG3* or adding CQ, induced *NNMT* mRNA and NNMT protein expression in H9c2 cells (Fig. 6A,B,E,F). Mitochondrial dysfunction was evidenced by reduced oxygen consumption rates (OCR) (Fig. 6D,G). If NAD$^+$ deficiency secondary to NNMT induction causes mitochondrial dysfunction, then silencing *NNMT* simultaneously with *ATG3* should restore mitochondrial function. Indeed, when *NNMT* was silenced simultaneously with *ATG3*, decreased OCR in *ATG3*-silenced cells was restored to levels similar to that in scrambled H9c2 cells (Fig. 6C,D). Also, product-inhibiting NNMT activity with exogenous MeNAM (Sen et al, 2019) in CQ-treated cells restored mitochondrial function to levels similar to that in control H9c2 cells (Fig. 6G). Of note, MeNAM alone did not affect mitochondrial function as evidenced by OCR (Appendix Fig. S5). Furthermore, inducing an NAD$^+$-deficient state by overexpressing NNMT in H9c2 cells (Fig. 5I) caused mitochondrial dysfunction (Fig. 6H) without changing autophagic flux. These data suggest that as in cAtg3-KO mouse hearts, blocking autophagic flux in H9c2 cells also induces NAD$^+$ deficiency by inducing NNMT, leading to increased methylation of NAM to MeNAM, and impaired mitochondrial function.

## Autophagic flux regulates NNMT protein levels

To confirm that autophagic flux alters NNMT protein expression, autophagic flux was induced by starving mice. In this experiment, autophagy induction significantly reduced NNMT protein and *NNMT* mRNA levels in the heart, which were blunted when *ATG3* was deleted (Fig. 6I,J). Other autophagic stimuli, such as ATG7 overexpression and Torin1 treatment, both of which increased autophagic flux also decreased NNMT protein expression in H9c2 cells (Fig. 6K,M), which was accompanied by decreased *NNMT* mRNA levels (Fig. 6L,N). These data reveal that autophagic flux regulates NNMT expression, identifying a novel regulatory circuit linking autophagy and NNMT expression in cardiomyocytes.

Given that autophagy could result in the engulfment of most cytosolic materials including RNA and proteins and target them to lysosomes for degradation (Guo et al, 2014; Liu et al, 2018), using an antibody against GFP, we isolated autophagosomes from H9c2 cells following adenoviral overexpression of GFP-LC3. These isolated autophagosomes contained ULK1, SQSTM1, and GFP-LC3-I and -II, and endogenous LC3-II, but not GAPDH or endogenous LC3-I. In the presence of CQ, isolated autophagosomes accumulated more ULK1, SQSTM1, and GFP-LC3-II and endogenous LC3-II, but not endogenous LC3-I, indicating that free LC3-I is not resident on autophagosomes. In cytosolic fractions, we observed ULK1, SQSTM1, GFP-LC3-I, endogenous LC3-I and LC3-II, and NNMT. Cytosolic SQSTM1, LC3-II, and NNMT were markedly increased in the presence of CQ. NNMT protein was not detected in isolated autophagosomes (Appendix Fig. S6A). Although GFP immunoprecipitates contained *NNMT* mRNA, GFP-LC3 immunoprecipitates did not pull down more NNMT mRNA in the absence or presence of CQ (Appendix Fig. S6B). These data suggest that autophagic regulation of NNMT expression is not secondary to selective autophagy.

## Blocking autophagy increases *NNMT* transcription by activating NF-κB signaling

We next investigated the mechanism by which autophagy deficiency enhances *NNMT* transcription. We focused on the nuclear factor kappa-light-chain-enhancer of activated B cells (NF-κB) in light of evidence that NF-κB may bind to the *NNMT* promoter to induce *NNMT* expression in skeletal myoblasts (Kim et al, 2010). NF-κB activation was measured using an NF-κB luciferase reporter in H9c2 cells in the presence of CQ. Inhibition of autophagy by CQ markedly increased NF-κB activation, which was inhibited by an NF-κB specific inhibitor Bay11-7085 (Bay) (Fig. 7A). To further confirm activation of NF-κB signaling following autophagy inhibition, the phosphorylation of RELA at S536 was measured revealing increased RELA phosphorylation in H9c2 cells following CQ exposure or *ATG3* silencing (Fig. 7B,C). These data were replicated in cAtg3-KO mouse hearts under fed conditions (Fig. 7D). When autophagic flux was induced by food deprivation in mouse hearts, RELA phosphorylation was significantly decreased, which was prevented in cAtg3-KO mouse hearts (Fig. 7D). Thus, inhibiting autophagy activates NF-κB signaling in cardiomyocytes.

We investigated if activated NF-κB signaling directly contributes to increased NNMT expression under conditions of autophagy deficiency. In mouse hearts, chromatin immunoprecipitation confirmed RELA binding to the *NNMT* promoter region (Fig. 7E), which was further enhanced when autophagic flux was blocked by CQ treatment (Fig. 7F). Increased *NNMT* expression in the presence of CQ was significantly inhibited by either *RELA* silencing or following incubation of cells with the NF-κB inhibitor Bay (Fig. 7G,H). Of note, the addition of Bay alone or silencing *RELA* alone did not alter *NNMT* mRNA expression. Thus,

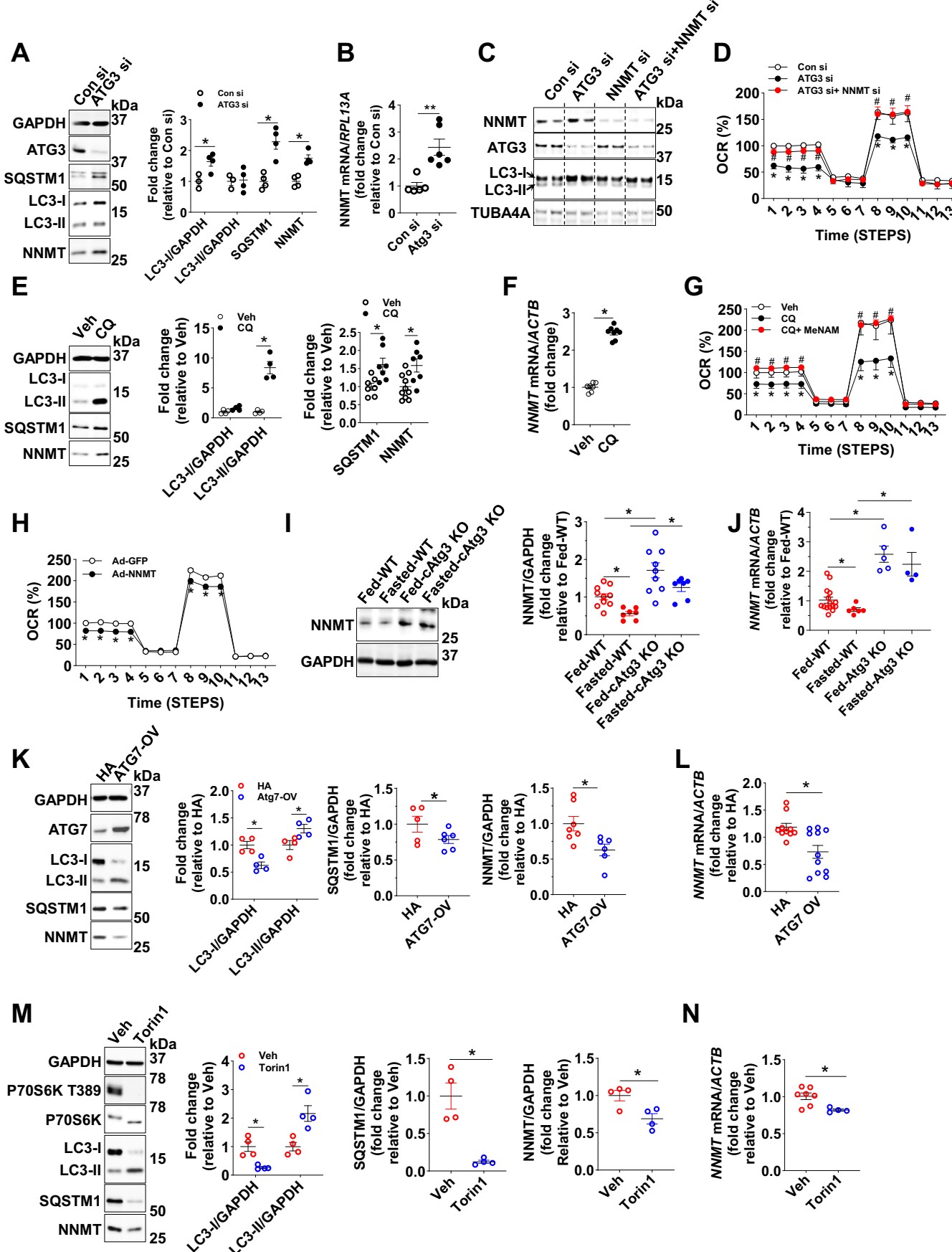

**Figure 6.  The manipulation of autophagic flux alters NNMT expression and thus affects mitochondrial function in cardiomyocytes.**

(A) Protein levels of GAPDH, ATG3, SQSTM1, LC3 (MAP1LC3A), and NNMT in H9c2 cells treated with *ATG3* siRNA (ATG3 si) or control siRNA (Con si). Representative blots are shown. $n = 4$ per group. Unpaired *t* tests were used to determine statistical significance between two groups. Data are mean ± SEM. *$P < 0.05$. (B) MRNA expression of *NNMT* in H9c2 cells treated with Atg3 si or Con si. $n = 5–6$ per group. Data are mean ± SEM. An unpaired *t* test was used to determine statistical significance between two groups. **$P < 0.01$. (C) Protein levels of NNMT, ATG3, LC3, and TUBA4A (alpha-tubulin) in H9c2 cells treated with ATG3 si, *NNMT* siRNA (NNMT si), or Con si, $n = 6$ per group. Representative blots are shown. (D) Oxygen consumption rate (OCR) in Con si-, ATG3 si-, or NNMT si-transfected H9c2 cells, $n = 5$ per group. Data are mean ± SEM. Unpaired *t* tests were used to determine statistical significance between two groups at corresponding time points. *$P < 0.05$ vs. Con si, #$P < 0.05$ vs. Atg3 si. (E) Protein levels of GAPDH, LC3, SQSTM1, and NNMT in H9c2 cells treated with Vehicle (Veh) or CQ at 20 µM for 12 h. Representative blots are shown. $n = 4–10$ per group. Unpaired *t* tests were used to determine statistical significance between two groups. Data are mean ± SEM. *$P < 0.05$. (F) *NNMT* mRNA levels in H9c2 cells treated with Veh or CQ at 20 µM for 12 h, $n = 7–8$ per group. Data are mean ± SEM. An unpaired *t* test was used to determine statistical significance between two groups. *$P < 0.05$. (G) OCR in H9c2 cells treated with Veh, CQ, or CQ + MeNAM. For OCR measurement, step 1–4: 20 µM glucose medium, step 5–7: oligomycin (1 µg/ml), step 8–10: FCCP (0.5 µg/ml), step 11–13: rotenone (1 µg/ml). $n = 4–5$ per group. Data are mean ± SEM. Unpaired *t* tests were used to determine statistical significance between two groups at corresponding time points. *$P < 0.05$ vs. Veh, #$P < 0.05$ vs. CQ. (H) OCR in H9c2 cells transfected with an adenovirus expressing GFP or NNMT. $n = 5–6$ per group. Data are mean ± SEM. Unpaired *t* tests were used to determine statistical significance between two groups at corresponding time points. *$P < 0.05$ vs. Ad-GFP. (I) NNMT and GAPDH protein levels in WT and cAtg3-KO mouse hearts. Mice, at ~16 weeks of age, were randomly fed or starved for 48 h, $n = 7–11$ per group. Data are mean ± SEM. One-way ANOVA followed by Bonferroni's multiple comparison tests was used to determine statistical significance. *$P < 0.05$. (J) *NNMT* mRNA levels in WT and cAtg3-KO mouse hearts, depicted in (I). $n = 4–15$ per group. Data are mean ± SEM. One-way ANOVA followed by Bonferroni's multiple comparison tests was used to determine statistical significance. *$P < 0.05$. (K) Protein levels of GAPDH, ATG7, LC3, SQSTM1, and NNMT in H9c2 cells transfected with a vector encoding HA or Atg7. $n = 4–7$ per group. Data are mean ± SEM. Unpaired *t* tests were used to determine statistical significance between two groups. *$P < 0.05$. (L) *NNMT* mRNA levels following ATG7 overexpression in same cells as (K). $n = 10–11$ per group. Data are mean ± SEM. An unpaired *t* test was used to determine statistical significance between two groups. *$P < 0.05$. (M) Protein levels of GAPDH, LC3, SQSTM1, P70S6K phosphorylation at T389, P70S6K, and NNMT in H9c2 cells treated with Torin1 at 10 µM for 12 h. $n = 4$ per group. Data are mean ± SEM. Unpaired *t* tests were used to determine statistical significance between two groups. *$P < 0.05$. (N) *NNMT* mRNA levels following Torin and CQ treatment in same cells as (M). $n = 4–8$ per group. Data are mean ± SEM. An unpaired *t* test was used to determine statistical significance between two groups. *$P < 0.05$. Source data are available online for this figure.

autophagy inhibition induces NNMT expression by activating NF-κB-regulated *NNMT* transcription in cardiomyocytes.

Next, we investigated if increased NNMT expression by activating NF-κB signaling under conditions of autophagy deficiency directly causes $NAD^+$ deficiency and mitochondrial dysfunction. In the presence of CQ, H9c2 cells exhibited $NAD^+$ deficiency and mitochondrial dysfunction as evidenced by decreased OCR, which were restored when Bay was added simultaneously (Fig. 7I,J). These data suggest that activated NF-κB signaling under conditions of autophagy deficiency in H9c2 cells causes $NAD^+$ deficiency and mitochondrial dysfunction in concert with NNMT induction.

## SQSTM1 accumulation secondary to autophagy inhibition activates NF-κB signaling

We next investigated the potential molecular mechanisms linking the inhibition of autophagy flux to the activation of NF-κB signaling. Studies have shown that SQSTM1 accumulation activates NF-κB signaling (Chen et al, 2016; Duran et al, 2008). SQSTM1 is dramatically accumulated when autophagic flux is blocked, suggesting SQSTM1 as a potential mediator. We, therefore, tested the hypothesis that SQSTM1 accumulation mediates autophagy deficiency-induced NF-κB activation and subsequent NNMT induction. To this end, we overexpressed SQSTM1 in H9c2 cells and observed increased RELA phosphorylation at S356 and NF-κB activation determined using an NF-κB-regulated luciferase reporter in the absence of changes in autophagic flux (Fig. 7K,L). In parallel, NNMT protein and *NNMT* mRNA expression were significantly increased (Fig. 7K,M), which was accompanied by $NAD^+$ deficiency and mitochondrial dysfunction (Fig. 7N,O). These data indicate that SQSTM1 accumulation under conditions of autophagy deficiency could contribute to NNMT induction and subsequent $NAD^+$ deficiency and mitochondrial dysfunction. To further evaluate this mechanism, we simultaneously silenced *SQSTM1* in H9c2 cells in which *ATG3* was silenced. *ATG3* silencing alone led

to SQSTM1 accumulation, NF-κB activation, increased NNMT expression, $NAD^+$ deficiency, and mitochondrial dysfunction, which was blunted when *SQSTM1* was silenced simultaneously (Fig. 8A–C). To further validate these in vitro data, we generated constitutive cardiac-specific *SQSTM1* knockout mice (cSQSTM1-KO) and measured NF-κB activation and NNMT protein expression. CSQSTM1-KO mouse hearts exhibited reduced RELA phosphorylation, lower NNMT protein expression, and increased $NAD^+$ levels (Fig. 8D,E). Taken together, these data support a model whereby autophagy flux controls SQSTM1 accumulation, which regulates NF-κB signaling and subsequent NNMT expression, which in turn regulates $NAD^+$ content leading to its deficiency and mitochondrial dysfunction, when autophagy flux is impaired.

## NNMT Inhibition restores $NAD^+$ levels and rescues cardiac dysfunction in cAtg3-KO mice

Having shown that NNMT induction contributes to $NAD^+$ deficiency in cAtg3-KO mouse hearts, we next investigated if NNMT inhibition could restore $NAD^+$ levels and ameliorate cardiac dysfunction in cAtg3-KO mice. To this end, we administered 5-Amino-1-methylquinolinium iodide, a specific small molecule NNMT inhibitor (NNMTi), to 30-week-old Atg3 KO and age-matched control mice (Neelakantan et al, 2018; Neelakantan et al, 2017). Following 3 days of NNMTi administration via intraperitoneal injection, the autophagic defect in cAtg3-KO mouse hearts persisted as evidenced by decreased LC3-II expression and SQSTM1 accumulation (Fig. 8F), but the deficient states of $NAD^+$ in cAtg3-KO mouse hearts were significantly reversed to the levels in WT mouse hearts (Fig. 8G). Under conditions of isoflurane anesthesia, heart rates (HR) were not affected by NNMTi supplementation (Fig. 8H). Notably, NNMTi treatment normalized fractional shortening in 30-week-old cAtg3-KO mice (Fig. 8I). These data support the conclusion that $NAD^+$ deficiency in cAtg3-KO mouse hearts is attributable to NNMT induction, which contributes to cardiac dysfunction that is associated with autophagy deficiency.

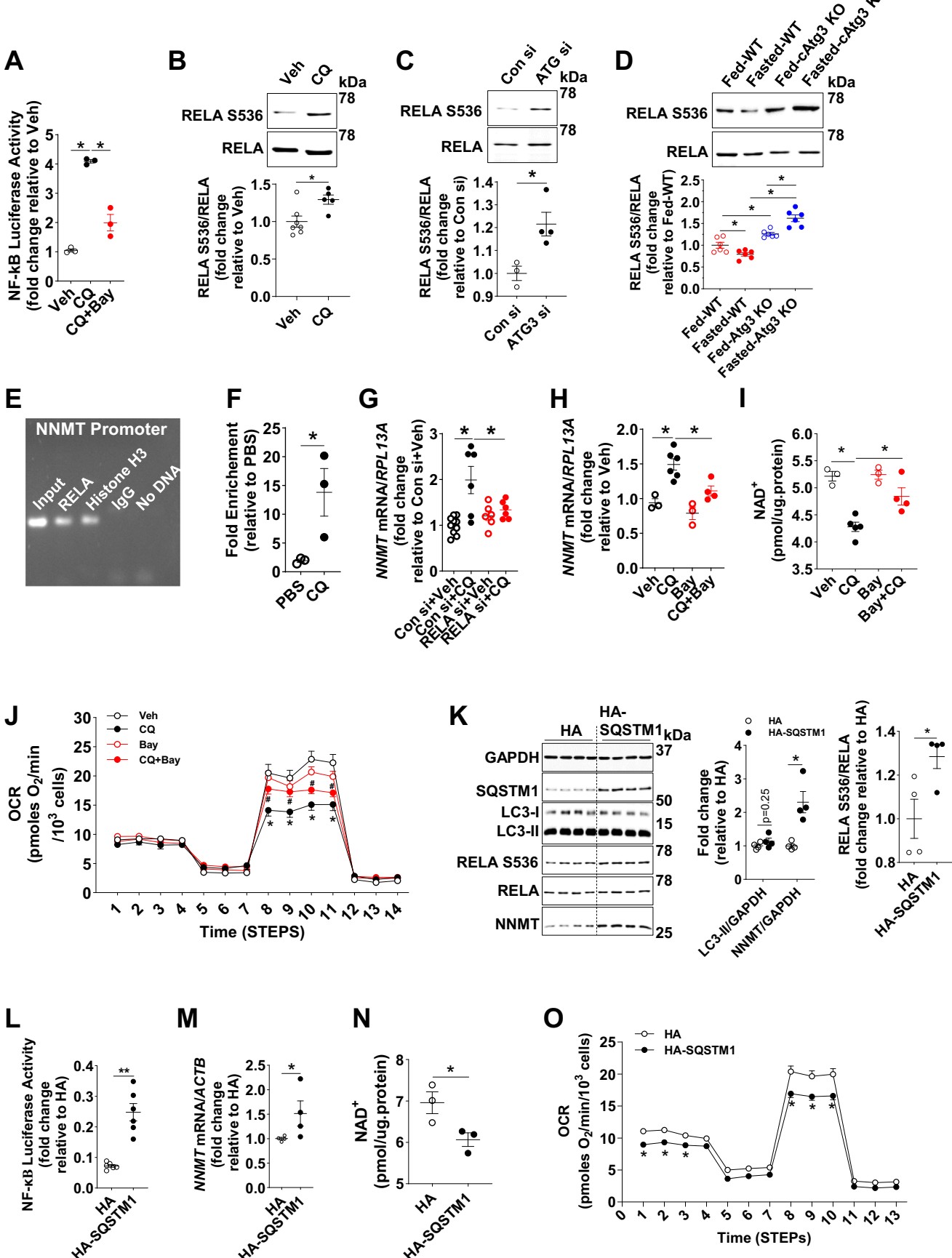

**Figure 7. Autophagy regulates NNMT transcription via NF-κB signaling.**

(A) NF-κB luciferase activity in H9c2 cells following treatment with PBS (Veh), CQ, or CQ + Bay11-7085 (Bay). H9c2 cells were first transfected with an NF-κB luciferase reporter vector or non-luciferase vector as a control for 24 h, and then were treated with PBS (Veh), CQ, or CQ + Bay for an additional 24 h in a nutrient-replete medium. $n = 3$ per group. Data are mean ± SEM. One-way ANOVA followed by Bonferroni's multiple comparison tests was used to determine statistical significance. *$P < 0.05$. (B) RELA (NF-κB p65) phosphorylation at S536 in H9c2 cells following treatment with Veh or CQ for 24 h, $n = 5$–7 per group. Data are mean ± SEM. An unpaired *t* test was used to determine statistical significance between two groups. *$P < 0.05$. (C) RELA phosphorylation at S536 in H9c2 cells following Atg3 knockdown, $n = 3$–4 per group. Data are mean ± SEM. An unpaired *t* test was used to determine statistical significance between two groups. *$P < 0.05$. (D) RELA phosphorylation at S536 in cAtg3-KO mouse hearts. Mice were randomly fed or fasted for 48 h, $n = 6$ per group. Data are mean ± SEM. One-way ANOVA followed by Bonferroni's multiple comparison tests was used to determine statistical significance. *$P < 0.05$. (E) Chromatin immunoprecipitation (CHIP) assay of RELA binding to the *NNMT* promoter region. Chromatin was extracted from WT mouse hearts, and protein-DNA immunoprecipitates were analyzed by RT-PCR, performed using primers specific to the NNMT promoter region. Histone H3 was used as positive control and IgG was used as a negative control. Representative images are shown. (F) Fold enrichment of RELA binding to the *NNMT* promoter in PBS or CQ-treated mouse hearts. $n = 3$ per group. Data are mean ± SEM. An unpaired *t* test was used to determine statistical significance between two groups. *$P < 0.05$. (G) *NNMT* mRNA expression in H9c2 cells. Cells were transfected with either Con si or *RELA* siRNA (RELA si), and then CQ was added to cells, $n = 6$–10 per group. Data are mean ± SEM. One-way ANOVA followed by Bonferroni's multiple comparison tests was used to determine statistical significance. *$P < 0.05$. (H) *NNMT* mRNA expression in H9c2 cells following treatment with Veh, CQ, Bay, or CQ + Bay for 24 h, $n = 3$ to 6 per group. Data are mean ± SEM. One-way ANOVA followed by Bonferroni's multiple comparison tests was used to determine statistical significance. *$P < 0.05$. (I) NAD+ in H9c2 cells following treatment with Veh, CQ, Bay, or CQ + Bay for 24 h, $n = 3$–5 per group. Data are mean ± SEM. One-way ANOVA followed by Bonferroni's multiple comparison tests was used to determine statistical significance. *$P < 0.05$. (J) Oxygen consumption rate (OCR) in H9c2 cells following treatment with Veh, CQ, Bay, or CQ + Bay for 24 h, $n = 6$–13 per group. Data are mean ± SEM. Unpaired *t* tests were used to determine statistical significance between two groups at corresponding time points. *$P < 0.05$ vs. Veh, #$P < 0.05$ vs. CQ. (K) Protein levels of GAPDH, LC3 (MAP1LC3A), SQSTM1, phosphorylation of RELA at S536, and NNMT; (L) NF-κB luciferase activity; (M) *NNMT* mRNA levels; and (N) NAD+ levels in H9c2 cells transfected with a vector encoding SQSTM1 or HA as control, $n = 4$–6 per group. Data are mean ± SEM. Unpaired *t* tests were used to determine statistical significance between two groups. *$P < 0.05$, **$P < 0.01$. (O) Oxygen consumption rate (OCR) in H9c2 cells transfected with a vector encoding SQSTM1 or HA as control, $n = 18$–22 per group. Data are mean ± SEM. Unpaired *t* tests were used to determine statistical significance between two groups at corresponding time points. *$P < 0.05$ vs. HA. Source data are available online for this figure.

# Discussion

In this study, we demonstrate that reducing autophagic flux accelerates NAM clearance leading to NAD+ deficiency, which results in mitochondrial and cardiac dysfunction in the heart. Specifically, impaired autophagic flux induces SQSTM1 accumulation that activates NF-κB signal transduction. The activated NF-κB binds the *NNMT* promoter to initiate *NNMT* transcription, which eventually increases NNMT protein content. As a key enzyme of NAD+ clearance, NNMT induction reduces NAD+ levels and antagonizes salvage synthesis of NAD+ from NAM, which precipitates mitochondrial and cardiac dysfunction.

Autophagic flux is a central homeostatic mechanism that maintains cellular homeostasis by degrading materials toxic to cardiomyocytes (Rubinsztein et al, 2011; Yamamoto et al, 2014). The detrimental effects of impaired autophagy on mitochondrial and cardiac function have been largely attributed to the accumulation of damaged proteins and organelles. Thus, most efforts have focused on why and how a defect in autophagic flux may occur under various pathophysiological conditions (Sciarretta et al, 2015; Zhang et al, 2017). Our study focused on identifying mechanisms arising from the impaired autophagic flux that could be independent of impaired cellular quality control in cardiomyocytes. By focusing on NAD+ metabolism in cardiomyocytes, we observed that impaired autophagic flux reduces NAD+ availability in cardiomyocytes, secondary to increased degradation, which underlies mitochondrial and cardiac dysfunction. Our data also supports a model in which autophagic flux via directional changes in SQSTM1 expression provides a novel regulatory circuit linking autophagy activation (usually induced by nutrient deficiency), with the regulation of NAD+, an important sensor of cellular nutrient status.

NAD+ functions as a cellular electron carrier in multiple metabolic pathways. It is also involved in other cellular processes, such as posttranslational modifications of proteins, for example, SIRT1-mediated deacetylation (Hsu et al, 2014). In failing hearts,

NAD+ levels decline and restoration of NAD+ has been reported to ameliorate cardiac dysfunction (Diguet et al, 2018; Matasic et al, 2018). NAD+ deficiency, therefore, has been considered one mechanism for heart failure (Yamamoto et al, 2014; Yano et al, 2015). NAD+ deficiency in failing heart tissues has been mainly attributed to decreased NAD+ salvage synthesis secondary, for example, to decreased expression of nicotinamide phosphoribosyl-transferase (NAMPT), a rate-limiting enzyme in the NAD+ salvage pathway that catalyzes the conversion of nicotinamide (NAM) to nicotinamide mononucleotide (NMN) (Diguet et al, 2018; Matasic et al, 2018), which is subsequently converted to NAD+ by nicotinamide mononucleotide adenylyl transferases (NMNATs) (Tatsumi et al, 2019). In addition, the activation of NADases, PARP and Sirtuin families, causes NAD+ deficiency in MEFs where autophagic flux is impaired (Kataura et al, 2022). Here, we show that in heart failure induced by reduced autophagic flux, NAD+ deficiency results from increased NAM clearance as a novel mechanism of decreased NAD+ salvage synthesis. We did not observe the changes in the expression of kinases that are involved in NAD+ salvage synthesis including NAMPT and NMNAT1-3 or NADases including Sirtuin families, PARP1/2, and BST1. Of note, protein expression of CD38, one of the NADases was decreased. Furthermore, we did not observe the changes in the activities of PARP1/2 and SIRT1. In contrast to heart failure from other causes, this NAD+ deficiency is due to induction of NNMT by SQSTM1-NF-κB signaling. Thus, although NAD+ deficiency plays a critical role in the development of heart failure in the mouse model of *ATG3* deficiency, which blocks autophagic flux, the mechanisms underlying NAD+ deficiency are distinct from other causes of heart failure. These data support the centrality of perturbed NAD+ metabolism in heart failure but reveal additional mechanisms by which this might occur.

Of note, the affinity of NNMT to NAM is ~400 μM, several orders of magnitude lower than that of NAMPT with $K_M$ of the low nanomolar range (Burgos and Schramm, 2008; Pissios, 2017), implying that the recycling of NAM back into NAD synthesis by

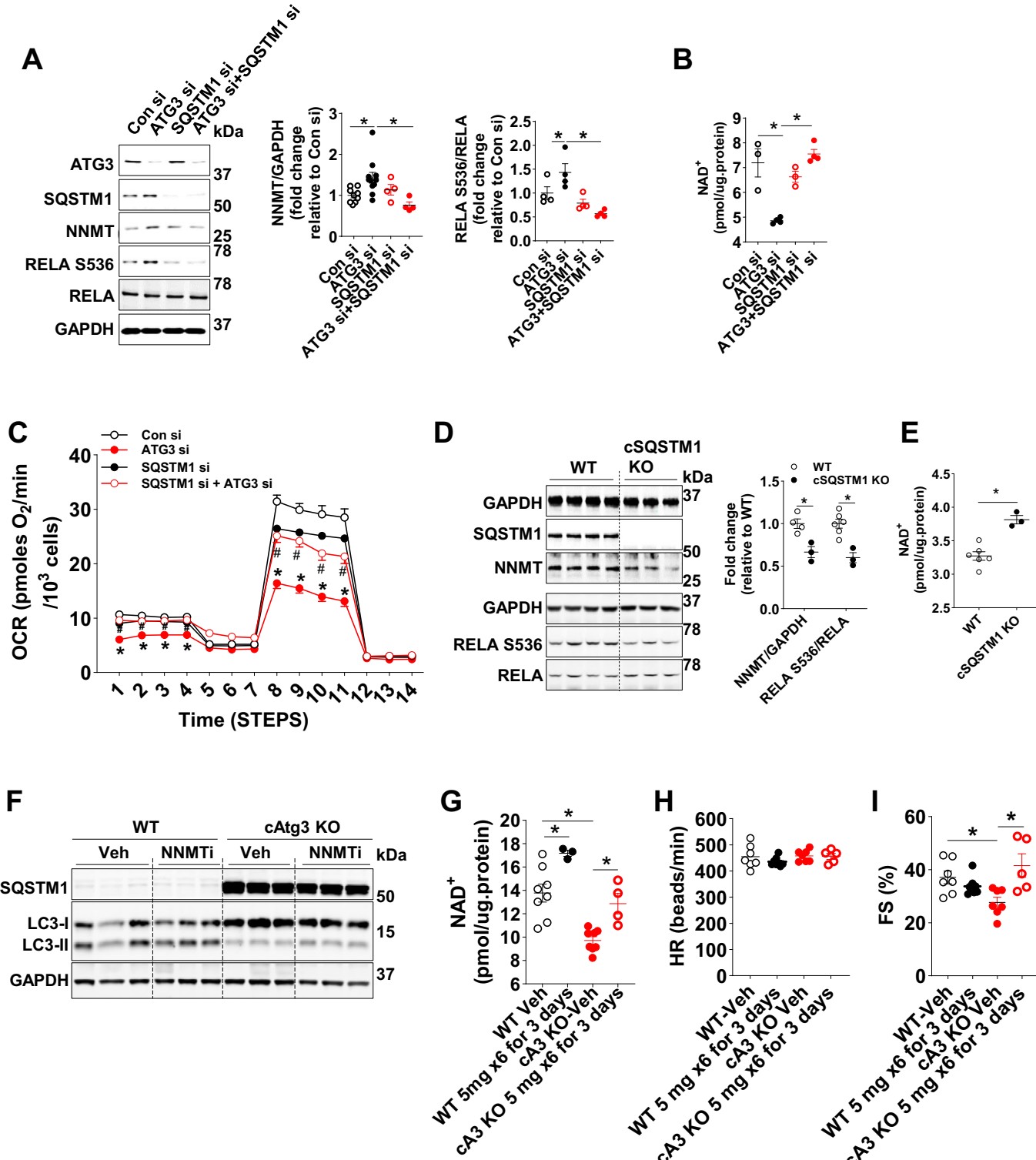

NAMPT may predominate over the formation of MeNAM via NNMT. It is therefore unexpected that the moderate upregulation of NNMT protein in the present study may outcompete NAMPT to methylate NAM toward MeNAM, resulting in NAD⁺ deficiency. One study has indicated that the coexistence of NNMT with NAMPT largely accelerates NAD⁺ consumption to result in NAD⁺ deficiency (Bockwoldt et al, 2019). However, the mechanisms by which NNMT competes with NAMPT to divert NAM toward MeNAM and whether NNMT could suppress NAMPT remains to be elucidated. Another interpretation is the release of NAM inhibition on NAD consumption enzymes to accelerate NAD⁺ consumption flux (Bockwoldt et al, 2019). However, in this study,

**Figure 8. SQSTM1 mediates NNMT transcription and inhibition of NNMT ameliorates cardiac dysfunction in cAtg3-KO mice.**

(A) Protein levels of ATG3, SQSTM1, NNMT, RELA (NF-κB p65) and RELA phosphorylation at S536, and GAPDH in H9c2 cells. *ATG3* and *SQSTM1* were silenced as indicated, $n = 4$–8 per group. Data are mean ± SEM. One-way ANOVA followed by Bonferroni's multiple comparison tests was used to determine statistical significance. *$P < 0.05$. (B) NAD$^+$ levels in H9c2 cells. *Atg3* and *SQSTM1* were silenced as indicated, $n = 3$–4 to 8 per group. Data are mean ± SEM. One-way ANOVA followed by Bonferroni's multiple comparison tests was used to determine statistical significance. *$P < 0.05$. (C) Oxygen consumption rate (OCR) in H9c2. *ATG3* and *SQSTM1* were silenced as indicated, $n = 9$–30 per group. Data are mean ± SEM. Unpaired $t$ tests were used to determine statistical significance between two groups at corresponding time points. *$P < 0.05$ vs. Con si, #$P < 0.05$ vs. Atg3 si. (D) Protein levels of GAPDH, SQSTM1, NNMT, RELA phosphorylation at S536, and RELA in cardiac-specific *SQSTM1* KO mouse hearts, $n = 3$–6 per group. Data are Mean ± SEM. Unpaired $t$ tests were used to determine statistical significance between two groups. *$P < 0.05$. (E) NAD$^+$ levels in cardiac-specific *SQSTM1* KO mouse hearts. $n = 3$–6 per group. Data are mean ± SEM. An unpaired $t$ test was used to determine statistical significance between two groups. *$P < 0.05$. (F) Protein levels of SQSTM1, LC3 (MAP1LC3A), and GAPDH in WT and cAtg3-KO mouse hearts. Mice, at ~30 weeks of age, were injected with 5-Amino-1-methylquinolinium iodide (NNMTi) or Veh as indicated. $n = 3$ per group. Typical blots are shown. (G) NAD$^+$ levels in WT and cAtg3-KO mouse hearts. Mice, at ~30 weeks of age, was injected with NNMTi or Veh as indicated. $n = 3$–8 per group. Data are mean ± SEM. One-way ANOVA followed by Bonferroni's multiple comparison tests was used to determine statistical significance. *$P < 0.05$. (H) Heart rates (HR) of WT and cAtg3-KO mice under isoflurane anesthesia. Mice, at ~30 weeks of age, were injected with NNMTi or Veh as indicated. $n = 5$–8 per group. (I) Fractional shortening (FS) in WT and cAtg3-KO mouse hearts under isoflurane anesthesia. Mice, at ~30 weeks of age, were injected with NNMTi or Veh as indicated. $n = 5$–8 per group. Data are mean ± SEM. One-way ANOVA followed by Bonferroni's multiple comparison tests was used to determine statistical significance. *$P < 0.05$. Source data are available online for this figure.

we did not observe increases in protein expression or activities of NAD$^+$-consuming enzymes. NNMT induction resulting in NAD$^+$ deficiency has also been observed in other tissues such as liver (Griffiths et al, 2021; Komatsu et al, 2018). Given the importance of NAD$^+$ metabolism in cellular homeostasis, it will be important in future studies, to illustrate how NNMT and NAMPT coordinately regulate NAD$^+$ consumption flux by diverting NAM.

Studies have shown that NAD$^+$ deficiency due to the inhibition of NAMPT blocks autophagic flux in cardiomyocytes, in part by SIRT1-mediated acetylation of autophagic mediators (Hsu et al, 2009; Hsu et al, 2014). In this study, we found that reduced autophagic flux may induce NAD$^+$ deficiency by accelerating NAM clearance without changing NAMPT protein expression. Thus, two distinct mechanisms of NAD$^+$ deficiency may cause cardiac dysfunction, which can be reversed by restoring the NAD$^+$ pool. Therefore, it is likely that autophagy and NAD$^+$ metabolism regulate each other, and the consequence or direction of this regulation might be context-dependent.

Autophagic flux has been reported to maintain mitochondrial structure and function by degrading dysfunctional mitochondria via mitophagy (Lee et al, 2012). Impaired autophagic flux has been associated with accumulation of dysfunctional mitochondria that may be characterized by impaired O$_2$ consumption, decreased ATP production, increased ROS generation and cytochrome *c* release that may precipitate apoptotic cell death (Nakai et al, 2007; Taneike et al, 2010; Wu et al, 2009). In this study, we observed decreased mitochondrial respiratory capacity that precedes cardiac dysfunction in ATG3-deficient hearts. Furthermore, we observed reduced mitochondrial content, which likely resulted in part from reduced mitochondrial biogenesis, given the reduced expression of transcriptional regulators of mitochondrial biogenesis such as PPARGC1A, NRF1, and NRF2. In addition, we observed evidence of inhibited mitophagy. Thus, impaired mitophagy and decreased mitochondrial biogenesis may simultaneously contribute to mitochondrial dysfunction when autophagic flux is impaired. Crosstalk between NAD$^+$ availability and PPARGC1A signaling, or the regulation of NAD$^+$ metabolic pathways by PPARGC1A have been described (Tran et al, 2016). What is clear, is that the well-documented relationships between NAD$^+$ metabolism and mitochondrial dysfunction (Kraus et al, 2014; Lee et al, 2019), or between autophagy and mitochondrial function is likely quite complex. A widely accepted mechanism is increased mitochondrial protein acetylation, attributed to NAD$^+$ deficiency and reduced

sirtuin activity (Horton et al, 2016). Increased acetylation of mitochondrial proteins has been implicated not only as a mechanism for perturbing mitochondrial energy homeostasis, but also for promoting mitochondrial MPTP opening (Karamanlidis et al, 2013). However, the relationship between PPARGC1A acetylation and mitochondrial biogenesis (Nemoto et al, 2005; Rodgers et al, 2005) is complex and incompletely understood, and whether or not acetylation of PPARGC1A alters its cellular functions is unclear. Thus, the precise mechanisms linking impaired autophagic flux or decreased NAD$^+$ availability with PPARGC1A expression or activity as observed in this study remains to be elucidated and will require additional study.

Many studies have demonstrated complex crosstalk between autophagy, mitochondria, and lipid metabolism (Civitarese et al, 2010; Dong and Czaja, 2011; Gumucio et al, 2019; Lee et al, 2013; Rambold et al, 2015; Schulze et al, 2016; Singh et al, 2009; Zhang et al, 2018), with impairment of any of these pathways perturbing the others. Thus, the coexistence of impaired autophagy, mitochondrial dysfunction, and impaired lipid metabolism often occurs, and is often accompanied by lipid accumulation (Civitarese et al, 2010; Dong and Czaja, 2011; Gong et al, 2015; Gumucio et al, 2019; Lee et al, 2013; Rambold et al, 2015; Schulze et al, 2016; Singh et al, 2009; Zhang et al, 2018). Consistent with this view, we observed altered fat metabolism in autophagy defective hearts, namely decreased fat oxidation in concert with evidence of myocardial lipid accumulation. Thus, ATG3-deficient hearts could be more susceptible to lipotoxicity. Although impaired mitochondrial metabolism could account for this phenomenon, the possibility also exists that depression of the NAD$^+$ pool might alter fat metabolism in other ways and will be the subject of future studies.

In conclusion, we describe a novel mechanism by which impaired autophagic flux induces mitochondrial dysfunction in cells and hearts and heart failure in vivo. Autophagic flux maintains cardiac and mitochondrial function by mediating SQSTM1-NF-κB-NNMT signal transduction that controls the cellular levels of NAD$^+$. Thus, the present study identifies a previously unrecognized regulatory pathway linking NAD$^+$ metabolism and autophagic flux. This pathway may be physiologically relevant linking nutrient deprivation, autophagy induction, and the maintenance of the cellular NAD$^+$ pool. Conversely, impaired autophagy, by promoting depletion of the NAD$^+$ pool, may contribute to tissue dysfunction such as heart failure. Consistent with previous studies (Diguet et al, 2018), this study provides strong experimental

evidence to further support that boosting NAD$^+$ metabolism represents a potential therapeutic strategy for ameliorating mitochondrial dysfunction and improving contractile function in heart failure (Diguet et al, 2018). Recent reports of small molecule NNMT inhibitors, which increase cellular NAD$^+$ levels leading to metabolic benefits such as the reversal of diet-induced obesity and prevention of muscle senescence (Neelakantan et al, 2019; Neelakantan et al, 2018; Sampson et al, 2021) supports the viability of modulating NNMT activity to achieve these goals.

# Methods

## Animal studies

The study protocol was approved by the Institutional Animal Care and Use Committees of the University of Utah, the Carver College of Medicine of the University of Iowa, and the David Geffen School of Medicine at the University of California, Los Angeles. Mice were maintained at the central animal facility with 12-h dark and 12-h light cycle, temperature at ~23 °C, and humidity at 40 to 60%. Mice had free access to water and food. All mice for experiments in this study were male on the C57BL6/J background. Female mice were used only for breeder cages and calculation of the percentage of cAtg3 mice (Appendix Table S1). Mice were starved for 5–6 h before sacrifice. Mice were anesthetized by chloral hydrate or 2% isoflurane gas with an inflow rate of 1 mL/min, and then hearts were immediately removed and rinsed in ice-cold phosphate-buffered saline (PBS, 10010023, ThermoFisher Scientific, Waltham, MA) before being snap-frozen in liquid nitrogen.

The $ATG3^{loxP/loxP}$ mice (Cai et al, 2018), $SQSTM1^{loxP/loxP}$ mice (Okada et al, 2009), and mice expressing Cre recombinase under the control of the α-myosin heavy chain promoter ($MYH6$-$Cre$) (Abel et al, 1999) were previously described. Constitutive cardiac-specific $ATG3$ KO mice were generated by crossing $ATG3^{loxP/loxP}$ mice with $MYH6$-Cre mice. Age-matched $ATG3^{loxP/loxP}$ mice without Cre were used as control mice. Constitutive cardiac-specific $SQSTM1$ KO mice were generated by crossing $SQSTM1^{loxP/loxP}$ mice with $MYH6$-$Cre$ mice. Age-matched $SQSTM1^{loxP/loxP}$ mice without Cre were used as controls.

Mice expressing a tamoxifen-inducible Cre recombinase under the control of the inducible cardiac-specific $MYH6$ promoter ($MYH6$-$MerCreMer$; B6.FVB(129)-A1cfTg($MYH6$-Cre/Esr1*)1Jmk/J) were obtained from The Jackson Laboratory (00565, Bar Harbor, ME). For inducible cardiac-specific $ATG3$ deletion, 6-week-old mice harboring $MYH6$-$MerCreMer$ and $ATG3^{loxP/loxP}$ genes were intraperitoneally injected with tamoxifen at a dose of 100 μg/g body weight/day for 5 consecutive days. Age-matched $ATG3^{loxP/loxP}$ mice without $MYH6$-$MerCreMer$ were injected with the same amount of tamoxifen as control mice.

## Cardiomyocyte isolation

Adult mouse cardiomyocytes were isolated as described in our previous study (Harris et al, 2022; Zhang et al, 2020). Briefly, adult mouse hearts were cannulated through the aorta and then perfused on a Langendorff system for ~5 min with Buffer A (113 mM NaCl, 4.7 mM KCl, 0.6 mM KH$_2$PO$_4$, 0.6 mM Na$_2$HPO$_4$, 1.2 mM MgSO$_4$, 10 mM HEPES, 12 mM NaHCO$_3$, 10 mM KHCO$_3$, 30 mM taurine, 10 mM 2,3-butanedione

monoxime [BDM; Sigma-Aldrich, B0753], 5.5 mM glucose, pH 7.0). The hearts were then perfused with a digestion buffer (Buffer A with the addition of Type II collagenase [Gibco, 17101015] at 300 U/mL and CaCl2 at 50 μM). The digestion times were determined by the hardness of the hearts and the rate of perfusion. Once the hearts were soft and the perfusion rate markedly increased, the ventricles were immediately dissected into small pieces in ice-cold Buffer A containing BSA (Research Products International, A30075) at 100 mg/mL. The cardiomyocyte suspension was then centrifuged at 2000×$g$ for 3 min at 4 °C. The supernatant was discarded, and the cardiomyocyte pellet was then washed twice by resuspension in ice-cold Buffer A containing BSA at 100 mg/mL and centrifugation for 3 min at 4 °C. After washing, cardiomyocyte pellets were homogenized in ice-cold King homogenization buffer (pH 7.2) containing 50 mM HEPES, 10 mM sodium pyrophosphate, 10 mM sodium fluoride, 2 mM EDTA, 2 mM sodium orthovanadate, 1% Triton X-100, and 10% glycerol, supplemented with Halt™ Protease and Phosphatase Inhibitor Cocktail, EDTA-free (78447, ThermoFisher Scientific) at a ratio of 1:100. Then lysates were centrifuged at 20,000×$g$ for 30 min at 4 °C, and the supernatants collected for immunoblotting.

## Transthoracic echocardiography

For assessment of LV fractional shortening, mice were anesthetized with 0.5–2% isoflurane gas with an inflow rate of 1 mL/min and placed on a heated stage (37 °C). Chest hair was then removed with a topical depilatory agent before echocardiography and pre-warmed ultrasonic gel was applied. B-mode images in parasternal long- and short-axis projections (at the papillary muscle level) were obtained using a Vivid 7 Pro ultrasound machine with a 13 MHz linear probe (GE Medical Systems, Boston, MA) by an experienced operator blinded to mouse identity. Fractional shortening (%) was calculated as [(LVDd − LVDs)/LVDd]*100, where LVDd is the LV dimension at diastole and LVDs is the LV dimension at systole (Riehle et al, 2011).

For assessment of LV ejection fraction (%), mice were sedated with Midazolam (0.3 mg/kg. Body weight) (NDC 0409-2305-17, Pfizer, New York, NY), and chest hair was then removed with a topical depilatory agent before the echocardiogram. Echocardiography was performed by an experienced operator blinded to mouse identity. M model images were obtained using a Vevo 2100 ultrasound machine with a 30 MHz linear array transducer (Visual Sonics, Bothell, Washington). Ejection fraction [%] was calculated as [(LVDd$^3$ − LVDs$^3$)/LVDd$^3$] * 100. Echocardiography data were analyzed by an operator in a blinded fashion.

## Histological analysis

Myocardial fragments were stained with H&E (047223.22, Thermo-Fisher Scientific; and HT110116, Sigma-Aldrich, St. Louis, MO) or Masson's trichrome (Sigma-Aldrich, HT15-1KT) as previously described (Riehle et al, 2013). Stained slides were analyzed as previously described (Mandarim-de-Lacerda, 2003).

## TUNEL staining

TUNEL staining was conducted in heart tissue using In Situ Cell Death Detection Kits (12156792910, Roche, Little Falls, NJ) following the supplier's protocol.

## Genomic DNA isolation and the measurement of DNA damage

Genomic DNA was isolated using a genomic DNA isolation kit (ab65358, Abcam, Waltham, MA), and the measurements of DNA damage were performed using a DNA damage-AP sites-Assay kit (ab211154, Abcam). The operations were completed according to the manufacturer's manuals.

## Oil red O staining

Heart samples were embedded in OCT on dry ice, and then sections were cut at ~8 μm thickness. After being fixed in 4% formalin in PBS, the sections were stained with Oil Red-O followed by lightly staining nuclei with hematoxylin (047223.22, ThermoFisher Scientific). The areas were randomly selected, and images were captured by bright-field microscopy.

## Starvation

Mice were deprived of food for 48 h starting at 8:00 AM, and then mice were immediately euthanized, and tissues harvested. Random-fed mice were used as controls.

## TGs measurement

TGs in heart tissues were measured using a Colorimetric Assay Kit (Cayman Triglyceride Colorimetric Assay Kit (10010303, Cayman, Ann Arbor, MI). The measurements were performed according to the manufacturer's manual.

## PARP activity assay

PARP activity was measured using a PARP/Apoptosis universal colorimetric assay kit (4677-096-K, R&D Systems, Minneapolis, MN), and the measurements were performed according to the manufacturer's manual.

## Assay of NAD$^+$ levels

NAD$^+$ levels in cells and heart tissues were measured using a NAD/NADH assay kit (ab6548, Abcam), and the measurements were performed according to the manufacturer's manual.

## Cell culture experiment

H9c2 cells (rat embryonic cardiomyoblasts; CRL-1446, ATCC, Manassas, VA) were maintained in high-glucose DMEM (11965-092, ThermoFisher Scientific) supplied with 10% FBS (97068-085, VWR, Radnor, PA). For siRNA knockdown of *Atg3*, *SQSTM1*, *NNMT*, and *RELA* (NF-κB p65), H9c2 cells were infected with siRNAs against *Atg3*, *SQSTM1* (6399, Cell Signaling Technology, Danvers, MA; or paired sequences as indicated), *NNMT* (L-101014-02-0005, GE Dharmacon), and RELA (6261S, Cell Signaling Technology), or scramble siRNA at 80–100 nM for 48–72 h together with Lipofectamine 2000 (11668500, Thermofisher Scientific) under nutrient-replete conditions. The sources of silencers or paired sequences are listed in Appendix Table S4.

For NNMT overexpression, H9c2 cells were infected with an adenovirus encoding GFP or NNMT (kind gift from Dr. Pavlos

Pissios, Beth Israel Deaconess Medical Center, Boston, MA; Appendix Table S5) (Hong et al, 2015), and further cultured in high-glucose DMEM with 10% FBS for an additional 48 h before harvest. The cells and medium were collected, respectively. The NAD$^+$ levels and its related metabolites were measured using LC-MS methods as described in previous studies (Trammell et al, 2016a, 2016b).

For SQSTM1 and ATG7 overexpression, H9c2 cells were infected with 5 μg/mL plasmid encoding *HA-FLAG* (10792, Addgene, Watertown, MA), *HA-SQSTM1* (28027, Addgene), or *ATG7* (24921, Addgene) for 60 h (Appendix Table S5) together with Lipofectamine 2000 (11668500, Thermofisher Scientific). Then, chloroquine (CQ, C6628, Sigma-Aldrich) at 20 μM was added to H9c2 cells for ~24 h. under nutrient-replete conditions.

For Torin1 (10997, Cayman) treatment, Torin1 at 100 nM was added to H9c2 cells for ~16 h under nutrient-replete conditions.

An NF-κB reporter kit (60614, BPS bioscience, San Diego, CA) was used for the measurement of NF-κB luciferase activity in cells. The experiment was performed according to the manufacturer's manual. Briefly, H9c2 cells were transfected with the NF-κB luciferase reporter vector or non-luciferase vector as control. After 24 h of transfection, H9c2 cells were treated with Veh (PBS) or CQ at 20 μM for an additional 24 h in the nutrient-replete medium. A dual-luciferase assay was performed based on the dual-luciferase reporter assay system (E1960, Promega, Madison, WI).

For cell starvation, H9c2 cells were washed with PBS twice, and cells were cultured in Hank's Balanced Salt Solution (HBSS, 1.26 mM CaCl$_2$, 0.5 mM MgCl$_2$, 0.4 mM MgSO$_4$, 0.4 mM KCl, 0.35 mM NaHCO$_3$, 8 mM NaCl, 0.05 mM Na$_2$HPO$_4$) supplemented with 0.5 mM pyruvate (P8574, Sigma-Aldrich).

## Autophagosome isolation

The protocol for autophagosome isolation from H9c2 cells was modified from protocols described in a previous report (Yao et al, 2019). In brief, H9c2 cells were first transfected with adenovirus encoding either GFP or GFP-tagged LC3, and then cells encoding GFP-LC3 were treated with either vehicle (Veh) or CQ at 20 μM for 16 h under nutrient-replete conditions. Cells were collected in the ice-cold separation buffer containing 250 mM sucrose (Sigma, S0389), 1 mM ethylenediaminetetraacetic acid (EDTA; E58100, Research Products International, Mt Prospect, IL), 10 mM 4-(2-hydroxyethyl)-1-piperazineethanesulfonic acid (HEPES; Research Products International, H750303), pH 7.4, with addition of Halt™ protease & phosphatase inhibitor cocktail at 1:1000 (ThermoFisher Scientific, 78446). Then, cell lysates were homogenized using a Dounce glass homogenizer, followed by centrifugation at 1000×*g* for 10 min at 4 °C to eliminate nuclei. The post-nuclear supernatant fractions were further centrifuged at 17,000×*g* for 20 min at 4 °C. Then, the supernatant fractions were collected as cytosolic fractions, and the pellet fractions containing autophagosomes were washed twice in PBS.

After washing, the pellet fractions were resuspended in the separation buffer, and primary antibody against GFP (55494, Cell Signaling Technology) was added to the pellet fractions, followed by incubation overnight at 4 °C. Next, protein A agarose beads (Millipore-Sigma, 16-125) were added and incubated for 1 h at 4 °C. The complexes were washed in ice-cold PBS for up to three or four times. Then, immunoblotting and qPCR were conducted, respectively.

For immunoblotting, following the addition of loading buffer (ThermoFisher Scientific, LC2676), the complexes were boiled at 85 °C for 5 min. Two controls, antibody (AB) no lysate and cytosolic fraction no AB were performed. For AB no lysate, the primary antibody against GFP was directly added to the separation buffer followed by the IP protocol as described above. For no AB, cytosolic fractions were boiled in loading buffer at 85 °C for 5 min. Immunoblotting was conducted as described below.

For qPCR, after washing, the pellet fractions were resuspended in TRizol reagent (ThermoFisher Scientific, 15596018), and total RNA in the pellet fractions was extracted and purified with the RNeasy kit (Qiagen, 74104). Quantitative real-time PCR was performed using SYBR Green (ThermoFisher Scientific, 2012607) with *RNU6-1* as a reference.

## Mitochondrial isolation

Mitochondria were isolated from fresh WT and cAtg3-KO mouse hearts. Fresh whole hearts were cut into pieces using a scissor in ice-cold buffer (pH 7.2) containing 70 mmol/L sucrose, 210 mmol/L D-mannitol, 1 mmol/L EGTA, 0.5% fatty acid-free BSA, and 5 mmol/L HEPES, and then homogenized using a Wheaton 903475 overhead Stirrer at speed of 8–10 for ~0.5 to 1 min on ice. The homogenates were then centrifuged at $4000 \times g$ for 4 min at 4 °C. After centrifugation, the pellets were discarded, and the supernatants were further centrifuged at $10,000 \times g$ for 10 min at 4 °C. The pellets were saved as the mitochondrial fraction, and the supernatant was saved as the cytosolic fraction. The mitochondrial fractions were then lysed in ice-cold King homogenization buffer (pH 7.2) containing 50 mM HEPES, 10 mM sodium pyrophosphate, 10 mM sodium fluoride, 2 mM EDTA, 2 mM sodium orthovanadate, 1% Triton X-100, and 10% glycerol, supplemented with Halt™ Protease and Phosphatase Inhibitor Cocktail, EDTA-free (78447, ThermoFisher Scientific) at a ratio of 1:100. Then lysates were then centrifuged at $20,000 \times g$ for 30 min at 4 °C, and the supernatants were collected for immunoblotting.

## Oxygen consumption analysis

Oxygen consumption was measured in H9c2 cells using a Seahorse XF Analyzer (Agilent Technology, Santa Clara, CA) as described before (Dott et al, 2014). H9c2 cells were seeded in Seahorse cell culture plates (Agilent Technology, Santa Clara, CA). Basal oxygen consumption rates (OCR) and extracellular acidification rates (ECAR) were measured to establish baseline rates. The cells were then treated with the following compounds in succession: 1 µg/mL oligomycin (Sigma, O4876), 0.5 µg/mL FCCP (Sigma, C2920), and 1 µg/mL rotenone (Sigma, R8875). Mitochondrial respirometry by Seahorse was conducted by an operator in a blinded fashion.

For the measurement of oxygen consumption in permeabilized cardiac fibers, left ventricular (LV) subendocardial muscle fibers were used to measure oxygen consumption rates and ATP synthesis as previously described (Boudina et al, 2005). Oxygen consumption rates were determined in the presence of palmitoyl-carnitine (Sigma, P1645) at 0.02 mM ($V_O$), 1 mM ADP (A2754, Sigma) ($V_{ADP}$), and 1 µg/mL of the ATP synthase inhibitor oligomycin (A7699, Sigma) ($V_{Oligo}$). ATP levels were measured in fibers after the addition of 1 mM ADP.

## Chromatin immunoprecipitation (CHIP) assay

ChIP assays were performed as described previously with minor modification (Gao et al, 2015). Chromatin was extracted from mouse hearts using the SimpleChIP kit according to the manufacturer's instructions (9004S, Cell Signaling Technology). Briefly, tissues were cross-linked with 1% formaldehyde (F10800, Research Products International) for 10 min and then lysed by micrococcal nuclease (EN0181, ThermoFisher Scientific) and sonicated using a sonic dismembrator five times for 15 s each time. RELA (NF-κB p65) protein was immunoprecipitated from precleared lysates with Protein A/salmon sperm DNA agarose beads (16-157, Millipore-Sigma,). DNA was released from protein-DNA complexes by proteinase K (115 879 001, Roche) digestion and then subjected to qPCR using power SYBR green kit. The RELA binding region of the *NNMT* promoter was amplified using the following primers: forward, 5′-TAG CCA TGA GCT CGT TTC CT-3′, and reverse, 5′-CAG AAG CCC TGA GTC CAG AG-3′. ChIP-qPCR data were normalized to total chromatin.

## Metabolomics analysis

Mice were euthanized, and hearts were immediately removed, rinsed in ice-cold PBS, and frozen in liquid nitrogen. Glycolytic and citric acid cycle metabolites in hearts were detected using GC-MS as previously described (McClain et al, 2013). This assay was conducted by an operator in a blinded fashion.

## In vivo NAD⁺ flux metabolomics

Mice were intraperitoneally injected with heavy-isotope-labeled nicotinamide riboside (NR) ($^{13}$C, D double-labeled NR, $^{13}$C, D-NR) as previously described (Trammell et al, 2016a, 2016b). 0, 30, and 60 min after intraperitoneal injection of $^{13}$C, D-NR, mice were euthanized, and hearts were harvested. For metabolomics analyses, hearts were immediately removed, freeze-clamped and rapidly submerged in liquid nitrogen. NAD$^+$ levels and its related metabolites were measured using LC-MS methods as described in previous studies (Trammell et al, 2016a, 2016b).

## NMN administration

β-nicotinamide mononucleotide (NMN; N3501, Sigma) was prepared in PBS and injected intraperitoneally to 3-week-old mice at 500 mg/kg/d for 7 days. PBS (pH 7.4) was injected intraperitoneally as controls. Mice were euthanized 6 h. after the last injection, and hearts were immediately removed, rinsed in ice-cold PBS, and then frozen in liquid nitrogen. ATP, ADP, AMP, NAD$^+$, and NADH were measured using LC-MS methods as previously reported (Shibayama et al, 2015).

## Administration of 5-amino-1-methylquinolinium iodide

5-amino-1-methylquinolinium iodide (NNMTi; 42464-96-0, Tocris, Minneapolis, MN) was first dissolved in DMSO and then resuspended in vehicle (0.2% CMC, 0.25% Tween 80, and 2% DMSO). The solution was sterilized through a 0.45-µm filter. NNMTi was injected intraperitoneally to mice at 5 mg/kg twice/day for 3 days. The vehicle (0.2% CMC, 0.25% Tween 80, and 2%

DMSO) was injected intraperitoneally as control. Twelve hours after the last injection of NNMTi, mice were subjected to echocardiography under isoflurane anesthesia as described in "Transthoracic echocardiography".

## Measurement of the activities of citrate synthesis (CS), carnitine palmitoyltransferase (CPT), glyceraldehyde 3-phosphate dehydrogenase (GAPDH), and Sirt1

For measurements of CS activity, hearts were homogenized in ice-cold homogenization buffer containing 20 mM HEPES (Research Products International) and 10 mM EDTA (Research Products International) at pH 7.4. Homogenates were subjected to two freeze/thaw cycles to liberate CS from the mitochondrial matrix and then diluted to an approximate final protein concentration of 1 µg/µL. The reaction was performed in 1 mL of reaction buffer containing 20 mM HEPES (H7523, Sigma), 1 mM EDTA, 220 mM sucrose (S0389, Sigma), 40 mM KCl (Sigma, P3911), 0.1 mM 5,5'-dithio-bis (2-nitrobenzoic acid) (DTNB, Sigma, D8130), and 0.1 mM acetyl-CoA (pH 7.4 at 25 °C) (A1625, Sigma). The reaction was started by the addition of 0.05 mM oxaloacetate (O4126, Sigma) and monitored at 412 nm for 3 min using an Epoch Microplate Spectrophotometer (BioTek Instruments, Inc. Winooski, VT). The result was normalized to protein content.

For CPT activity measurement, mitochondria were isolated from fresh heart tissue as previously described (Riehle et al, 2011). Mitochondria were assayed in 1 mL of reaction buffer containing 20 mM HEPES (H7523, Sigma), 1 mM EGTA (E3889, Sigma), 220 mM sucrose (S0389, Sigma), 40 mM KCl (P3911, Sigma), 0.1 mM DTNB (D8130, Sigma), 1.3 mg/ml BSA, and 40 µM palmitoyl-CoA (P9716, Sigma). The reaction was started by the addition of 1 mM carnitine (C0283, Sigma) and monitored at 412 nm for an additional 4 min using an Epoch Microplate Spectrophotometer (BioTek Instruments). The CPT activity was normalized to mitochondrial protein levels.

For measurements of GAPDH and SIRT1 activities, heart tissues were homogenized and GAPDH and SIRT1 activities were measured according to the manufacturer's manuals (ab204732 and ab156065, Abcam). Exogenous NAD+ (10127965001, Sigma-Aldrich) was added to the reaction buffer to a final concentration of 10 mM.

## Administration of colchicine (Col) and chloroquine (CQ) to mice

For Col (C9754, Sigma) treatment, Col was dissolved in PBS and injected intraperitoneally in mice at 18, 6, and 2 h before heart harvest at doses of 4, 1, and 1 µg/g body weight, respectively. For CQ treatment, CQ was dissolved in PBS and injected intraperitoneally in mice at 48, 24, and 2 h. before heart harvest at doses of 30, 30, and 50 µg/g body weight, respectively. After the treatments above, mice were euthanized, hearts were immediately removed and rinsed in ice-cold PBS, and then frozen in liquid nitrogen.

## Quantitative real-time PCR

Total RNA was extracted from hearts or H9c2 cells using TRIzol reagent and purified with the RNeasy kit. Quantitative real-time PCR was performed using SYBR Green with ROX as an internal reference dye. The expression level was normalized to the transcript levels of the *RPL13A*, *GAPDH*, *RNU6-1*, or *ACTB*. The primer sequences are listed in Appendix Table S2.

## Immunoblotting

For immunoblotting, proteins were resolved on SDS-PAGE and analyzed using the LI-COR Odyssey Imager (LI-COR). The following primary antibodies were used (Appendix Table S3): ATG3 (A3231), MAP1LC3A (LC3, L8918), TUBA4A (alpha-Tubulin, T5168) from Sigma-Aldrich; ACTB (beta-Actin, 3700), SQSTM1 (p62, 5114), ATG7 (8858), Acetylated-Lysine (9441), FOXO1 (2880), GAPDH (2118), RELA S536 (NF-κB p65 S536, 3036), RELA (NF-κB p65, 6956), MTOR (2971), MTOR S2448 (4517), AMPKα T172 (2531), AMPKα (2793), P70S6K T389 (9206), P70S6K (2708), ULK1 (8054), Sirtuin antibody sampler kit (9787), SIRT4 (69786), PARP1 (9532), GFP (55494), Poly/Mono-ADP Ribose (83732), and PRKN (Parkin, 4211) from Cell Signaling Technology; Ac-FKHR (D-19, sc-49437), PPARGC1A (PGC1α, H-300, sc-13067), GAPDH (SC-32233), SIRT1 (B-10, sc-74504), and TOMM20 (TOM20, sc-17764) antibodies from Santa Cruz Biotechnology (Santa Cruz Biotechnology, Dallas, TX); CD38 (60006-1-lg), BST1 (CD157, 16337-1-AP), NMNAT1 (11399-1-AP), NMNAT3 (13261-1-AP), PARP2 (2055-1-AP), and NNMT antibody (15123-1-AP) and PINK1 antibody (23274-1-AP) from Proteintech (Rosemont, IL); NMNAT2 (PA5-115662) from Invitrogen (ThermoFisher Scientific); SDHA (ab14715) antibody from Abcam.

## Statistical analysis

Statistical analyses were performed using GraphPad Prism 8. All data are presented as mean ± SEM. Statistical significance ($P < 0.05$) was determined by one-way ANOVA followed by Bonferroni's multiple comparison test. An unpaired $t$ test was used to compare between two groups.

# Data availability

All primary data presented in this study is available upon reasonable request. This study includes no data deposited in external repositories. No large-scale data amenable to repository deposition were generated in this study.

# Peer review information

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

## Acknowledgements

Studies were supported by NIH grants (R01HL108379, U54 HL112311) to EDA who is an established investigator of the American Heart Association. EDA was supported by the John B Stokes Chair in Diabetes Research from the Fraternal Order of Eagles Diabetes Research Center (000000). CB was supported by the Roy J. Carver Trust, R01HL147545 and the Alfred E. Mann Family Foundation (000000). SB was supported by NIH grant R01HL149870-01A1. QL was supported by AHA award 20CDA35320042. We thank Dr. Roberta A. Gottlieb (Cedars Sinai, UCLA, Los Angeles, CA) for providing the GFP-LC3 adenovirus. The GFP-adenovirus and NNMT-adenovirus were a kind gift from Dr. Pavlos Pissios (Beth Israel Deaconess Medical Center, Boston, MA, USA). We thank Dr. Timothy Graham for giving Atg3^loxP/loxP mice (Division of Endocrinology, Metabolism and Diabetes, School of Medicine, University of Utah, Salt Lake City, UT, USA). We thank Dr. Toru Yanagawa (Faculty of Medicine, University of Tsukuba, Ibaraki, Japan) for providing *SQSTML^loxP/loxP* mice. We are grateful for the support of the metabolomics core at the University of Utah, Salt Lake City, UT, USA.

## Author contributions

**Quanjiang Zhang**: Conceptualization; Formal analysis; Validation; Investigation; Writing—original draft; Writing—review and editing. **Zhonggang Li**: Conceptualization; Formal analysis; Validation; Investigation; Writing—original draft; Writing—review and editing. **Qiuxia Li**: Conceptualization; Formal analysis; Validation; Investigation; Writing—original draft; Writing—review and editing. **Samuel AJ Trammell**: Formal analysis; Investigation; Methodology. **Mark S Schmidt**: Investigation; Methodology. **Karla Maria Pires**: Investigation; Methodology. **Jinjin Cai**: Formal analysis; Investigation; Methodology. **Yuan Zhang**: Formal analysis; Investigation; Methodology. **Helena Kenny**: Formal analysis; Investigation; Methodology. **Sihem Boudina**: Conceptualization; Data curation; Formal analysis. **Charles Brenner**: Conceptualization; Data curation; Formal analysis. **E Dale Abel**: Conceptualization; Resources; Data curation; Formal analysis; Supervision; Funding acquisition; Validation; Visualization; Methodology; Writing—original draft; Project administration; Writing—review and editing.

## Disclosure and competing interests statement

CB is the chief scientific advisor of ChromaDex and co-founder of Alphina Therapeutics. The remaining authors declare no competing interests.

# Expanded View Figures

**A**

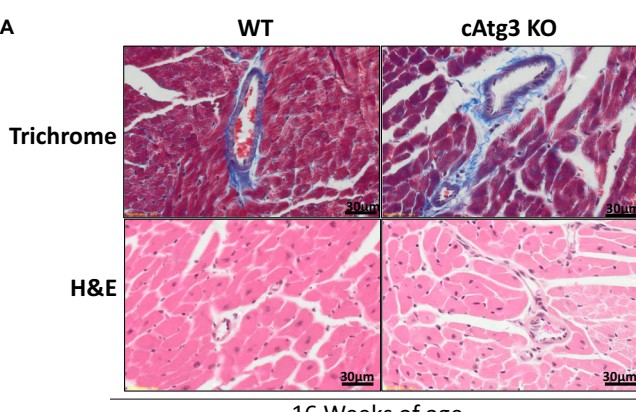

**B**

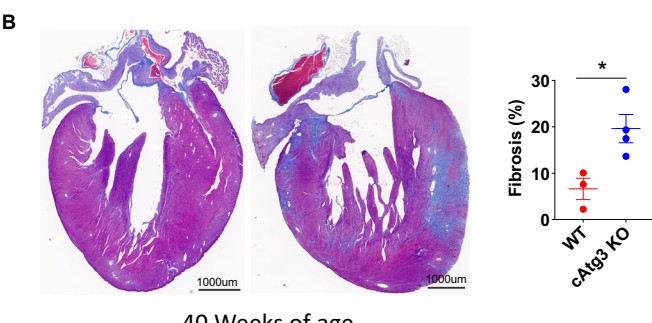

**Figure EV1. CAtg3-KO develops fibrosis in the heart with aging, related to Fig. 1.**

(**A**) Histology (Trichrome and Hematoxylin and Eosin (H&E)) of WT and cAtg3-KO mouse hearts at 16 weeks of age, $n = 6$ per group. Representative images are shown. (**B**) Histology (Trichrome and Hematoxylin and Eosin (H&E)) of WT and cAtg3-KO mouse hearts at 40 weeks of age, $n = 3$–4 per group. Data are mean ± SEM. An unpaired $t$ test was used to determine statistical significance between two groups. *$P < 0.05$. Source data are available online for this figure.

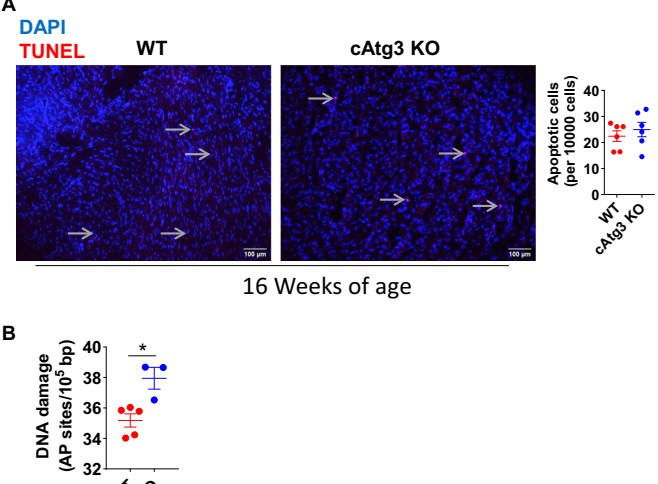

**16 Weeks of age**

**40 Weeks of age**

**Figure EV2. Apoptosis and DNA damage in cAtg3-KO with aging, related to Fig. 1.**

(**A**) TUNEL staining in WT and cAtg3-KO mouse hearts. Sections were obtained from mice at 16 weeks of age, $n = 6$ per group. Arrows represent TUNEL-positive nuclei. Data are mean ± SEM. (**B**) DNA damage in WT and cAtg3-KO mouse hearts at 40 weeks of age. $n = 3$–5 per group. Data are mean ± SEM. An unpaired $t$ test was used to determine statistical significance between two groups. *$P < 0.05$. Source data are available online for this figure.

**A**

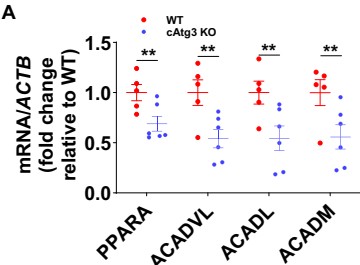

**B**

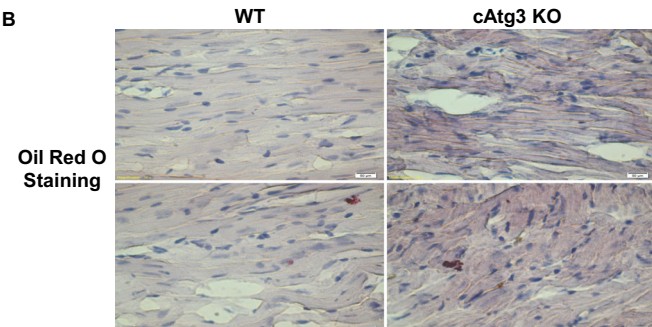

**Figure EV3. Decreased expression of fatty acid oxidation genes and increased lipid accumulation in cAtg3-KO hearts, related to Fig. 2.**

(**A**) mRNA levels of fatty acid oxidation related genes in 16-week-old WT and cAtg3-KO mouse hearts. $n = 4$–6. Data are mean ± SEM. Unpaired *t* tests were used to determine statistical significance between two groups. **$P < 0.01$. (**B**) Representative oil red O staining of tissue sections from WT and cAtg3-KO mouse hearts at 16 weeks of age, $n = 2$ per group. Representative section images are shown. Source data are available online for this figure.

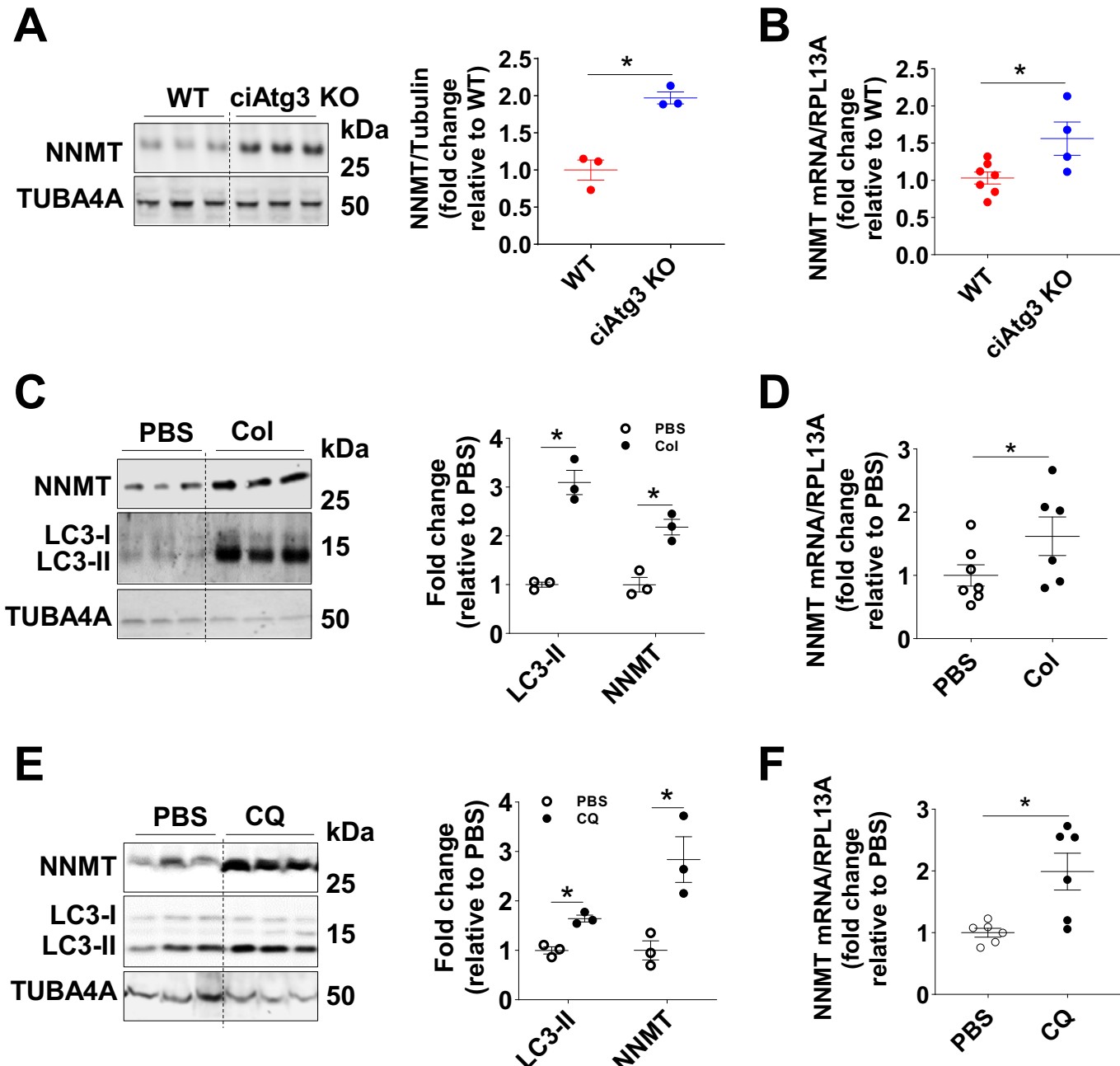

**Figure EV4. CAtg3-KO upregulates NNMT expression in hearts, related to Fig. 5.**

(A) NNMT protein levels in WT and ciAtg3 KO mice 1 week after last tamoxifen injection. $n = 3$ per group. Data are mean ± SEM. An unpaired $t$ test was used to determine statistical significance between two groups. *$P < 0.05$. (B) *NNMT* mRNA levels in WT and ciAtg3 KO mice 1 week after last tamoxifen injection. $n = 4$–7 per group. Data are mean ± SEM. An unpaired $t$ test was used to determine statistical significance between two groups. *$P < 0.05$. (C) NNMT and LC3 (MAP1LC3A) protein levels in hearts of WT mice following treatment with PBS, or Colchicine (Col), $n = 3$ per group. Data are mean ± SEM. An unpaired $t$ test was used to determine statistical significance between two groups. *$P < 0.05$. (D) *NNMT* mRNA levels in hearts of WT mice following treatment with PBS, or Colchicine (Col). $n = 6$–7 per group. Data are mean ± SEM. An unpaired $t$ test was used to determine statistical significance between two groups. *$P < 0.05$. (E) NNMT and LC3 protein levels in hearts of WT mice following treatment with PBS, or Chloroquine (CQ). $n = 3$ per group. Data are mean ± SEM. An unpaired $t$ test was used to determine statistical significance between two groups. *$P < 0.05$. (F) *NNMT* mRNA levels in hearts of WT mice following treatment with PBS, or CQ. $n = 6$ per group. Data are mean ± SEM. An unpaired $t$ test was used to determine statistical significance between two groups. *$P < 0.05$. Source data are available online for this figure.

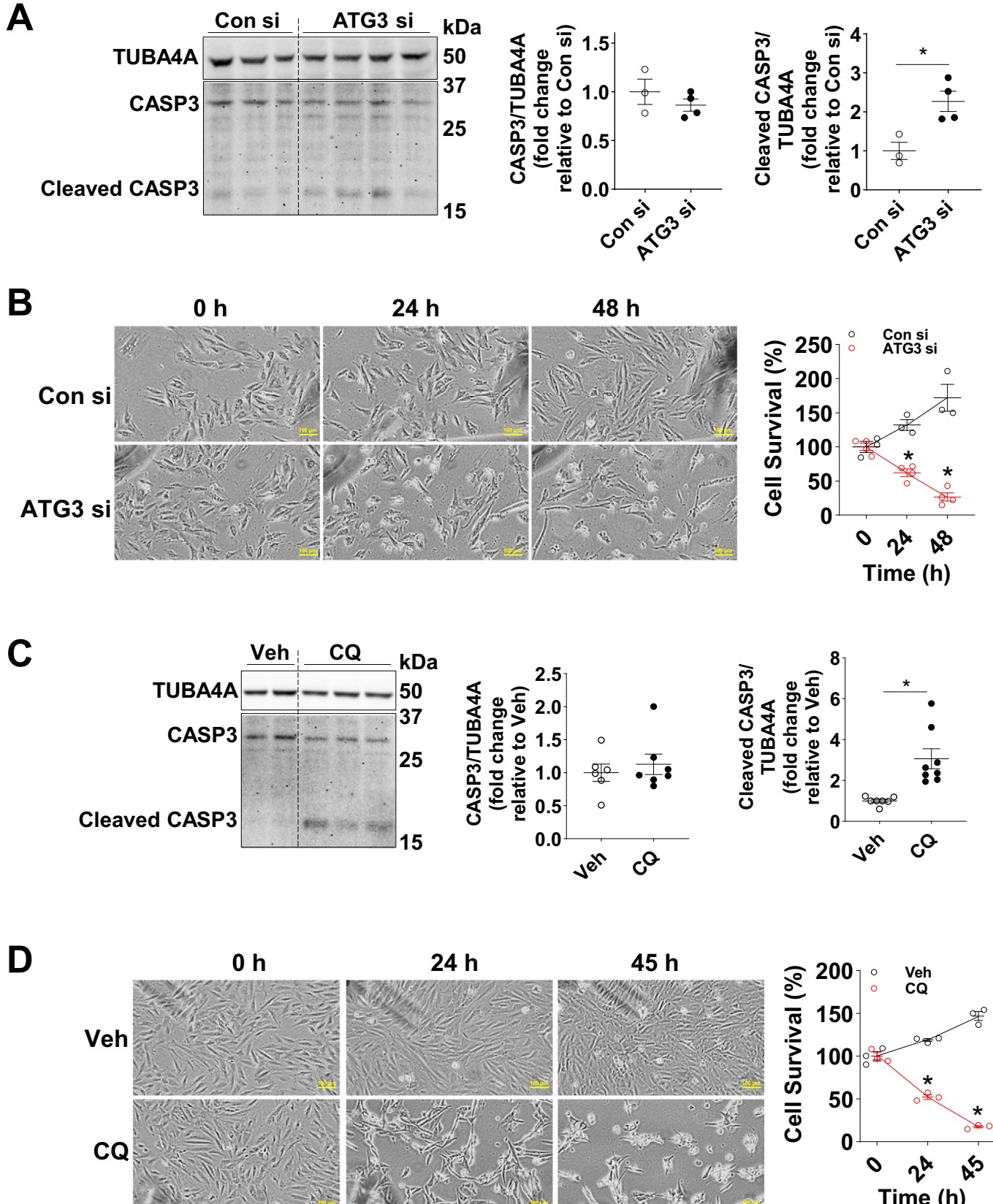

◀  **Figure EV5.  Cell death under conditions of defective autophagy, related to Fig. 6.**

(**A**) Protein levels of CASP3 (Caspase3), Cleaved CASP3, and TUBA4A (alpha-tubulin) in H9c2 cells transfected with *ATG3* siRNA. *ATG3* siRNA was added to cells for ~48 h prior to harvesting cells for western blot. $n = 3$–4 per group. Data are mean ± SEM. Unpaired $t$ tests were used to determine statistical significance between two groups. *$P < 0.05$. (**B**) Cell death in H9c2 cells cultured in nutrient-replete medium. H9c2 cells were transfected with *ATG3* siRNA for ~48 h. Then, the cells were moved to the growth medium (high-glucose DMEM supplemented with 10% FBS), and the images in the same location in each well were captured at the end of siRNA transfection (T0) and then at 24 h and 48 h thereafter. $n = 3$–4 images at each time point per group. Data are mean ± SEM. Unpaired $t$ tests were used to determine statistical significance between two groups at corresponding time points. *$P < 0.05$ vs. Con si-treated H9c2 cells at the corresponding time points. (**C**) Protein levels of CASP3, Cleaved CASP3, and TUBA4A in H9c2 cells treated with 20 µM chloroquine (CQ) for 16 h. $n = 6$–8 per group. Data are mean ± SEM. Unpaired $t$ tests were used to determine statistical significance between two groups. *$P < 0.05$. (**D**) Cell death in H9c2 cells cultured in nutrient-replete medium. H9c2 cells were treated with 20 µM CQ. There was no evidence of cell death at 16 h. However, cell viability is reduced at 24 h. Images of H9c2 cells in the same location in each well were captured at time points indicated, followed by cell counting. $n = 3$ images at each time point per group. Data are mean ± SEM. Unpaired $t$ tests were used to determine statistical significance between two groups at corresponding time points. *$P < 0.05$ vs. Veh-treated H9c2 cells at the corresponding time points. Source data are available online for this figure.

