## [Peer Review File · The EMBO Journal]

Control of NAD⁺ homeostasis by autophagic flux modulates mitochondrial and cardiac function

quanjiang zhang, Zhonggang Li, Qiuxia Li, Samuel Trammell, Mark Schmidt, Karla Pires, Jinjin Cai, Yuan Zhang, Helena Kenny, Sihem Boudina, Charles Brenner, and E. Abel

DOI: [10.15252/emboj.2023113557](https://doi.org/10.15252/emboj.2023113557)

Corresponding author: E. Abel (dabel@mednet.ucla.edu)

Review Timeline:

Submission Date:	20th Jan 23
Editorial Decision:	12th Mar 23
Revision Received:	18th Sep 23
Editorial Decision:	19th Oct 23
Revision Received:	31st Oct 23
Accepted:	8th Nov 23

Editor: Daniel Klimmeck

Transaction Report:

Dear Dr Dale Abel,

Thank you again for the submission of your manuscript (EMBOJ-2023-113557) to The EMBO Journal. Please accept my sincere apologies for getting back to you with unusual protraction due to delayed referee input, as well as detailed discussion in the editorial team. Your study was assessed by three reviewers with expertise in autophagy and metabolism, whose comments are enclosed below.

As you will see from their comments, the referees acknowledge the analysis and potential interest and value of your findings on p62-NFkB-NNMT-dependent NAD⁺ clearance as a key mechanism explaining. However, they also express major concerns. i.p. referee #1 states substantial issues regarding clinical relevance of the findings (ref#1, pt.1) as well as subtlety of the effects and proof for a causal involvement of NNMT in autophagy loss-induced mitochondrial dysfunction and heart failure (ref#1, pt.2: cross commenting ref#3). Further, referee #2 is concerned about data inconsistencies which need clarification in his/her view (ref#2, pts.3,4). Reviewer #3 requests corroboration of NAD⁺ level dependence by additional experiments testing autophagy inhibition in wild-type mice (ref#3, pt.2).

Given the overall interest stated and broader angle of your findings, we are able to invite you to revise your manuscript experimentally to address the referees' comments, along the lines sketched in your outline. I need to stress though that we do require strong support from the referees on a revised version of the study in order to move on to publication of the work. As to the open outcome of the revisional work I suggest keeping EMBO Reports in mind for this study as an alternative venue.

We have discussed i.p. the request for additional genetic support of NNMT involvement and concluded that this experiment, while per se well taken, is beyond a reasonable revision for the current study. Please note however that given the outcome of the described inhibitor experiments is entirely open at this point, yet the evidence required to address a key concern of the referees, we consider this a threshold case at this point. Accordingly, I suggest keeping EMBO Reports in mind for this study as an alternative venue.

Please feel free to contact me if you have any questions or need further input on the referee comments.

When submitting your revised manuscript, please carefully review the instructions below.

Please feel free to approach me any time should you have additional questions related to this.

Thank you for the opportunity to consider your work for publication.

I look forward to your revision.

Best regards,

Daniel Klimmeck

Daniel Klimmeck, PhD
Senior Editor
The EMBO Journal

Instruction for the preparation of your revised manuscript:

- 1) a .docx formatted version of the manuscript text (including legends for main figures, EV figures and tables). Please make sure that the changes are highlighted to be clearly visible.
- 2) individual production quality figure files as .eps, .tif, .jpg (one file per figure).
- 3) a .docx formatted letter INCLUDING the reviewers' reports and your detailed point-by-point response to their comments. As part of the EMBO Press transparent editorial process, the point-by-point response is part of the Review Process File (RPF),

which will be published alongside your paper.

4) a complete author checklist, which you can download from our author guidelines ([https://wol-prod-cdn.literatumonline.com/pb-assets/embo-site/Author Checklist%20-%20EMBO%20J-1561436015657.xlsx](https://wol-prod-cdn.literatumonline.com/pb-assets/embo-site/Author%20Checklist%20-%20EMBO%20J-1561436015657.xlsx)). Please insert information in the checklist that is also reflected in the manuscript. The completed author checklist will also be part of the RPF.

6) It is mandatory to include a 'Data Availability' section after the Materials and Methods. Before submitting your revision, primary datasets produced in this study need to be deposited in an appropriate public database, and the accession numbers and database listed under 'Data Availability'. Please remember to provide a reviewer password if the datasets are not yet public (see <https://www.embopress.org/page/journal/14602075/authorguide#datadeposition>).

7) Our journal encourages inclusion of *data citations in the reference list* to directly cite datasets that were re-used and obtained from public databases. Data citations in the article text are distinct from normal bibliographical citations and should directly link to the database records from which the data can be accessed. In the main text, data citations are formatted as follows: "Data ref: Smith et al, 2001" or "Data ref: NCBI Sequence Read Archive PRJNA342805, 2017". In the Reference list, data citations must be labeled with "[DATASET]". A data reference must provide the database name, accession number/identifiers and a resolvable link to the landing page from which the data can be accessed at the end of the reference. Further instructions are available at .

8) At EMBO Press we ask authors to provide source data for the main and EV figures. Our source data coordinator will contact you to discuss which figure panels we would need source data for and will also provide you with helpful tips on how to upload and organize the files.

Numerical data can be provided as individual .xls or .csv files (including a tab describing the data). For 'blots' or microscopy, uncropped images should be submitted (using a zip archive or a single pdf per main figure if multiple images need to be supplied for one panel). Additional information on source data and instruction on how to label the files are available at .

9) We replaced Supplementary Information with Expanded View (EV) Figures and Tables that are collapsible/expandable online (see examples in <https://www.embopress.org/doi/10.15252/emboj.201695874>). A maximum of 5 EV Figures can be typeset. EV Figures should be cited as 'Figure EV1, Figure EV2' etc. in the text and their respective legends should be included in the main text after the legends of regular figures.

11) For data quantification: please specify the name of the statistical test used to generate error bars and P values, the number (n) of independent experiments (specify technical or biological replicates) underlying each data point and the test used to calculate p-values in each figure legend. The figure legends should contain a basic description of n, P and the test applied. Graphs must include a description of the bars and the error bars (s.d., s.e.m.).

Please remember: Digital image enhancement is acceptable practice, as long as it accurately represents the original data and

conforms to community standards. If a figure has been subjected to significant electronic manipulation, this must be noted in the figure legend or in the 'Materials and Methods' section. The editors reserve the right to request original versions of figures and the original images that were used to assemble the figure.

We realize that it is difficult to revise to a specific deadline. In the interest of protecting the conceptual advance provided by the work, we recommend a revision within 3 months (10th Jun 2023). Please discuss the revision progress ahead of this time with the editor if you require more time to complete the revisions.

Referee #1:

In the present manuscript the authors claim that "impaired autophagy induced mitochondrial dysfunction and heart failure is due to an kFKB mediated induction of NNMT and the consumption of NAM that would impair NAD homeostasis. Although somewhat interesting the manuscript lacks in novelty and also the data does not fully support the conclusions that NNMT increase is the culprit of the NAD-decline.

Major concerns:

- 1) Figures 1 and 2 are not extremely novel and are expected based on data that supports the role of autophagy in heart function. The actual clinical significance is not clear here since there is no data indicating that ATG3 plays a role in human cardiac dysfunction. Furthermore, A full analysis of the cardiac parameters would be important. Is this systolic only or diastolic also.
- 2) The main issues is that the effects on NAM and methyl-NAM are small compared with the effect on NAD the main there are many technical issues with the models and presentation of the work and major issues with providing convincing support for the premise that NNMT is the cause of NAD decline and if NAD-decline is indeed drive by the increase of NNMT. It would be necessary to cross the ATG 3 KO with the NNMT KO mouse to demonstrate. This is in the view of this reviewer a necessary experiment to prove the claim of the study that NNMT is the cause of the NAD decline, it appears that much more validation would be needed to substantiate this claim, and determination of the other components of NAD homeostasis need to be done in detail not only expression of the mRNA.

Referee #2:

In this manuscript Zhang, Li et al demonstrate an NAD deficit in the heart with autophagy deficiency and investigate mechanisms underlying this phenotype. The authors propose a cascade whereby accumulation of p62 results in the activation of NF-kB, expression of NNMT and increased NAD clearance. Overall, the conclusions are based on strong data in vivo and in vitro, the analyses are done rigorously, conclusions are balanced, and there is a lot of conceptual novelty in the dataset. The manuscript also provides further support to the connection between autophagy deficiency to NAD depletion, both considered hallmarks of aging. However, before the manuscript can be recommended for publication several points need to be addressed as outlined below.

1. Data at 16-weeks of age suggests lack of fibrosis or apoptosis in cAtg3 KO mice. Autophagy deficiency in the heart has previously been shown to result in these phenotypes (e.g. <https://doi.org/10.4161/auto.6.5.11947>) and it would be helpful if the authors provided an explanation, whether this still occurs in cAtg3 KO but at the later stage, either by showing data or through discussion.
2. Fig 2H-K indicates impaired mitochondrial biogenesis. Could the authors speculate where this defect could originate from, e.g. is it a direct effect of autophagy deficiency or could impaired mitochondrial turnover via autophagy be coupled to reduced biogenesis?
3. Fig. EV5 indicates lack of sirtuin or PARP activation. The data on sirtuins is in agreement with the published data that NAD⁺ level is a limiting factor for sirtuin activity. However, it is puzzling that PARP activity (which could contribute to NAD depletion in addition to accelerated clearance by NNMT) is unchanged, considering an established link between autophagy deficiency and DNA damage accumulation (<http://dx.doi.org/10.1016/j.tcb.2016.11.011>). It would be interesting to see PARylation levels and investigate DNA damage markers in this model.
4. There are some inconsistencies in the paper, where some readouts are shown for one dataset but not others. E.g. NNMT mRNA levels in Fig. EV7 are shown for some but not other experimental comparisons, and it would be good to show it for ciAtg3 KO and CQ. Similarly, mRNA levels for NNMT in Atg3 KO (Fig. 6I).
5. Fig. 6 - Do experiments with autophagy and NF-kB inhibition in H9c2 cells also lead to NAD⁺ changes, e.g. is reduced OCR due to lower NAD⁺ levels or is it a direct effect of NNMT? Is overexpression of p62 sufficient to deplete NAD and reduce OCR? Is depletion of p62 in Atg3si cells sufficient to rescue OCR deficit? It would be good to improve cohesion between the steps of the proposed cascade by showing NAD levels and mitochondrial activity for some of these conditions.

6. The first paragraph of Discussion summarises findings on the effect of autophagy dysfunction in the heart mediated by NF- κ B. However, all the NF- κ B data is based on cell culture experiments. As before, it would improve consistency if NF- κ B activation was shown in heart tissue with autophagy deficiency. Similarly, p62 KO (Fig. 7M) could link better to the rest of the story if NF- κ B and NAD data in this model was provided.

Minor points:

1. Page 4 "Atg3, a key mediator of autophagic initiation that mediates the conversion of LC3-I to LC3-II" needs reference.
2. Page 5 first paragraph last sentence requires reference.

Referee #3:

Impaired autophagy has been shown to cause mitochondrial dysfunction and heart failure. While altered mitophagy and protein quality control contribute to, it is not known whether other mechanisms involved in impaired autophagic flux could explain mitochondrial dysfunction and heart failure. In this study, the authors show that Atg3 deficiency-induced impaired autophagy led to accelerated NAD⁺ clearance and NAD⁺ deficiency. This contributes to the development of mitochondrial and cardiac dysfunction and resultant heart failure.

Overall, this is a comprehensive, well documented study. The authors used several models and approaches to decipher the mechanisms. The mainly evaluated the components of the NAD⁺ pathway on mRNA level, protein levels and some activities. They also used beside Atg3-deficient mice, experiments in H9c2 cells to test the consequences of activation (torin 2) or inhibition (chloroquine) of autophagy on the OCR or NNMT, as well as luciferase assay for NF- κ B. The authors proposed that lower NAD⁺ levels in autophagy inhibition result from the activation of the p62-NF- κ B-NNMT signal transduction.

Major comments

1. The authors claim that the absence of conversion of LC3-I to -II indicates impaired autophagic flux. It would be nice to validate whether inhibition of the autophagic flux in wild-type mice (e.g. with leupeptin) produces similar effects on increased NAD⁺ clearance rate than in Atg3-deficient mice.
2. The ratio of LC3-II/LC3-I is not accepted by the Autophagy guidelines. Therefore, it would be better to relate LC3-II on GAPDH only.
3. Several data, particularly for the pathway analysis (Figure 4 and 5) are done on variable number of mice (for example Figure 5A: n=7,6, then 8 for WT and n=5,6 and 5 for cAtg3KO mice). I would be more secure in my review to see the same number of mice in all panels.
4. The loading control GAPDH for detection of NNMT is not appropriate, because of very close MW. Therefore, I have doubt that both were done on the same membrane shown in Figures 6 and 7. Full blots would be good in the supplemental file.
5. Figure 7F: would be nice to see the CHIP in this condition.
6. Figure 7M: is the cp62 KO associated with higher autophagy activity?
7. Do you have any evidence of cell death in vitro (Atg3si or chloroquine)?
8. Is mitophagy defective in Atg3KO mice?

Minor comments

1. EV4: representative graph - correct NANMAT1-3 by NMNAT1-3.
2. MW markers should be shown in all blots, including the EV.
3. In the introduction some references to other paper showing autophagy alteration in human or mouse models of inherited cardiomyopathy would be nice.

Responses to Referee #1's comments:

We appreciate these insightful comments. These comments provided us the opportunity to improve the present study. We completed additional experiments to address the reviewer's comments.

In the present manuscript the authors claim that "impaired autophagy induced mitochondrial dysfunction and heart failure is due to an kFKB mediated induction of NNMT and the consumption of NAM that would impair NAD homeostasis. Although somewhat interesting the manuscript lacks in novelty and also the data does not fully support the conclusions that NNMT increase is the culprit of the NAD-decline.

Major concerns:

Q1: 1) Figures 1 and 2 are not extremely novel and are expected based on data that supports the role of autophagy in heart function. The actual clinical significance is not clear here since there is no data indicating that ATG3 plays a role in human cardiac dysfunction. Furthermore, A full analysis of the cardiac parameters would be important. Is this systolic only or diastolic also.

Response: We agree that an observation linking autophagy impairment to cardiac dysfunction per se is not novel. Indeed, there is a significant body of literature linking impaired autophagy with heart failure. The novelty of the work is that cardiac dysfunction under the condition of autophagy impairment is attributable to increased NAD catabolism. In addition, the cause of autophagy impairment in human heart failure is multifactorial, thus a single gene knockout does not represent the complexity of the multiple pathophysiological mechanisms that lead to heart failure. Thus, the relevance of this work is mechanistic, linking autophagy impairment with NAD metabolism. We provide data in hearts, but also in cultured cells, indicating that this mechanism is likely cell-autonomous.

Reduced ejection fraction (EF) and fractional shortening (FS) are echocardiographic parameters that estimate systolic function. Systolic dysfunction in concert with induction of BNP is consistent with heart failure with reduced ejection fraction. As such, we do not believe that specific measures of diastolic performance, although potentially of interest, would contribute specifically to our focus on the mechanisms responsible for NAD reduction in the context of autophagy impairment, that correlates with ventricular dysfunction.

Q2: 2) The main issue is that the effects on NAM and methyl-NAM are small compared with the effect on NAD the main there are many technical issues with the models and presentation of the work and major issues with providing convincing support for the premise that NNMT is the cause of NAD decline and if NAD-decline is indeed drive by the increase of NNMT. It would be necessary to cross the ATG 3 KO with the NNMT KO mouse to demonstrate. This is in the view of this reviewer a necessary experiment to prove the claim of the study that NNMT is the cause of the NAD decline, it appears that much more validation would be needed to substantiate this claim, and determination of the other components of NAD homeostasis need to be done in detail not only expression of the mRNA.

Response: Thank you for this insightful comment. The authors accept that in terms of the heart failure phenotype, it would be ideal to cross ATG3 and NNMT floxed mice to generate double floxed animals that would then be used to generate cardiac-specific KOs of ATG3 and NNMT. Given that NNMT KO mice are not immediately available, we have instead used pharmacological NNMT inhibition to address this question. The small molecule NNMT inhibitor (NNMTi, 5-Amino-1-methylquinolinium iodide) has been used in mouse experiments

(Neelakantan *et al*, 2018a; Neelakantan *et al*, 2017), where they have been shown, to inhibit NNMT and reverse metabolic disturbances associated with NAD deficiency. We therefore administered this NNMTi to 30-week-old cAtg3 KO mice and measured cardiac function. As shown in Fig 8F to I of the revised manuscript, the administration of NNMTi normalized NAD levels in cAtg3 KO mice and rescued cardiac dysfunction. These additional data, support our hypothesis and demonstrate that NNMT induction mediates the decline in NAD, which contributes to cardiac dysfunction in cAtg3 KO mouse hearts. This *in vivo* data supports our data in cultured cells demonstrating impaired mitochondrial bioenergetics after atg3 silencing that is rescued by silencing NNMT, supporting a cell-autonomous mechanism linking autophagy impairment, with impaired mitochondrial energetics occurring as a consequence of an NNMT-dependent increase in NAD catabolism (Figure 6D).

Neelakantan H, Vance V, Wetzel MD, Wang HL, McHardy SF, Finnerty CC, Hommel JD, Watowich SJ (2018b) Selective and membrane-permeable small molecule inhibitors of nicotinamide N-methyltransferase reverse high fat diet-induced obesity in mice. *Biochem Pharmacol* 147: 141-152

Neelakantan H, Wang HY, Vance V, Hommel JD, McHardy SF, Watowich SJ (2017) Structure-Activity Relationship for Small Molecule Inhibitors of Nicotinamide N-Methyltransferase. *J Med Chem* 60: 5015-5028

Responses to Referee #2's comments:

We thank this referee for his/her insightful comments and suggestions. These comments provided us the opportunity to improve the present study. We completed additional experiments to address the reviewer's comments.

Referee #2:

In this manuscript Zhang, Li *et al* demonstrate an NAD deficit in the heart with autophagy deficiency and investigate mechanisms underlying this phenotype. The authors propose a cascade whereby accumulation of p62 results in the activation of NF- κ B, expression of NNMT and increased NAD clearance. Overall, the conclusions are based on strong data *in vivo* and *in vitro*, the analyses are done rigorously, conclusions are balanced, and there is a lot of conceptual novelty in the dataset. The manuscript also provides further support to the connection between autophagy deficiency to NAD depletion, both considered hallmarks of aging. However, before the manuscript can be recommended for publication several points need to be addressed as outlined below.

Q1: 1. Data at 16-weeks of age suggests lack of fibrosis or apoptosis in cAtg3 KO mice. Autophagy deficiency in the heart has previously been shown to result in these phenotypes (e.g. <https://doi.org/10.4161/auto.6.5.11947>) and it would be helpful if the authors provided an explanation, whether this still occurs in cAtg3 KO but at the later stage, either by showing data or through discussion.

Response: We thank the reviewer for this comment. Indeed, at 16 weeks of age, cAtg3 KO mouse hearts do not exhibit fibrosis or apoptosis, which contrasts with previous studies in which other autophagy proteins were reduced in the heart (e.g. <https://doi.org/10.4161/auto.6.5.11947>). We stained mouse heart sections of WT and cAtg3 KO mice at approx. 40 weeks of age when cAtg3 KO mice exhibit reduced ejection fraction by echocardiography in lightly sedated mice (See methods in the revised manuscript). As shown in Fig EV2B, cAtg3 KO mouse hearts exhibit increased in fibrosis. In addition, DNA damage is

also increased, consistent with increased apoptotic cell death in cAtg3 KO mouse hearts as they age.

Q2: 2. Fig 2H-K indicates impaired mitochondrial biogenesis. Could the authors speculate where this defect could originate from, e.g. is it a direct effect of autophagy deficiency or could impaired mitochondrial turnover via autophagy be coupled to reduced biogenesis?

Response: We agree with the reviewer that mitochondrial biogenesis is impaired based on reduced levels of mitochondrial proteins in concert with reduced PGC1 α . The basis for the reduction in PGC1 α protein likely reflects transcriptional repression as evidenced by reduced PGC1 α mRNA. The mechanisms for the transcriptional repression are not known, but we discuss potential mechanisms in the revised manuscript. The reviewer's comment also prompted us to ascertain if there could be other mechanisms that could contribute to reduced mitochondrial capacity in these animals, that could be exacerbated by an inadequate adaptive response exemplified by reduced PGC1 α expression. As such, we evaluated mitophagy in WT vs cAtg3 KO mouse hearts. As shown in Fig 2H and I, mitophagy is impaired in cAtg3 KO mouse hearts, indicating impaired mitochondrial turnover. Together, these data have identified an additional mechanism that could impair mitochondrial integrity in addition to impaired biogenesis.

Q3: 3. Fig. EV5 indicates lack of sirtuin or PARP activation. The data on sirtuins is in agreement with the published data that NAD⁺ level is a limiting factor for sirtuin activity. However, it is puzzling that PARP activity (which could contribute to NAD depletion in addition to accelerated clearance by NNMT) is unchanged, considering an established link between autophagy deficiency and DNA damage accumulation (<http://dx.doi.org/10.1016/j.tcb.2016.11.011>). It would be interesting to see PARylation levels and investigate DNA damage markers in this model.

Response: We thank this referee for this interesting comment. We evaluated PARylation and DNA damage in WT vs cAtg3 KO mouse hearts. As shown in Fig EV7C, PARylation is not altered in WT vs cAtg3 KO mouse hearts at the age of 16-weeks, implying that PARP is not activated in this model at time point when NAD⁺ levels are already reduced. Consistent with increased fibrosis at 40 weeks of age, DNA damage is significantly increased (Fig EV3B), which could be consistent with increased apoptosis in cAtg3 KO mouse hearts as they age. However, we note a reduction in NAD⁺ in young Atg3KO hearts months before fibrosis occurs. Moreover, our studies in cultured cells occur on a relatively short time scale, which indicates that the relationship between NAD catabolism and autophagy on the basis of NNMT induction could lead to NAD depletion without the need to invoke PARP activation in response to DNA damage. Taken together, these new data strengthen our conclusions that PARP activation does not contribute to NAD⁺ depletion in Atg3KO mouse hearts.

Q4: 4. There are some inconsistencies in the paper, where some readouts are shown for one dataset but not others. E.g. NNMT mRNA levels in Fig. EV7 are shown for some but not other experimental comparisons, and it would be good to show it for ciAtg3 KO and CQ. Similarly, mRNA levels for NNMT in Atg3 KO (Fig. 6I).

Response: We thank this referee for this suggestion. NNMT mRNA is now shown in fed and fasted Atg3KO mice (Fig. 6J), in ciAtg3 KO hearts, and in animals treated with colchicine or chloroquine (Fig. EV8).

Q5: 5. Fig. 6 - Do experiments with autophagy and NF-kB inhibition in H9c2 cells also lead to NAD⁺ changes, e.g. is reduced OCR due to lower NAD⁺ levels or is it a direct effect of NNMT? Is overexpression of p62 sufficient to deplete NAD and reduce OCR? Is depletion of p62 in Atg3si cells sufficient to rescue OCR deficit? It would be good to improve cohesion between the steps of the proposed cascade by showing NAD levels and mitochondrial activity for some of these conditions.

Response: We agree that this is a reasonable concern. The data of OCR and NAD in H9c2 cells treated with CQ in the absence or presence of Bay 11-7085 has been added (Fig 7I and J). This new data indicates that CQ treatment significantly decreases NAD⁺ levels and OCR, which is blocked in the presence of Bay 11-7085. New data on OCR and NAD⁺ in H9c2 cells in which p62 is overexpressed has been added (Fig 7N and O). These new data indicates that p62 overexpression decreases NAD⁺ levels and OCR. New data of OCR and NAD in H9c2 cells in which Atg3 and p62 were silenced simultaneously is also included. These new data indicate that Atg3 knockdown significantly decreases NAD⁺ levels and OCR, which is prevented when p62 is silenced simultaneously (Fig 8B).

Q6: 6. The first paragraph of Discussion summarises findings on the effect of autophagy dysfunction in the heart mediated by NF-kB. However, all the NF-kB data is based on cell culture experiments. As before, it would improve consistency if NF-kB activation was shown in heart tissue with autophagy deficiency. Similarly, p62 KO (Fig. 7M) could link better to the rest of the story if NF-kB and NAD data in this model was provided.

Response: We thank this reviewer for these insightful comments. New data on NF-kB p65 phosphorylation in WT vs cAtg3 KO mouse hearts has been added in Fig. 7D. This new data indicates that cAtg3 KO mouse hearts show increased NF-kB p65 phosphorylation compared to WT mouse hearts. The data of NF-kB p65 phosphorylation and NAD⁺ levels in WT vs cp62 KO mouse hearts have been added in Fig. 8D and E. These new data indicates that cp62-KO mouse hearts show decreased NF-kB p65 phosphorylation and NAD⁺ levels compared to WT mouse hearts.

Minor points:

Q7: 1. Page 4 "Atg3, a key mediator of autophagic initiation that mediates the conversion of LC3-I to LC3-II" needs reference.

Response: The references have been added.

Q7: 2. Page 5 first paragraph last sentence requires reference.

Response: The references have been added.

Responses to Referee #3's comments:

We thank this referee for his/her insightful comments and suggestions. These comments provided us the opportunity to improve the present study. We completed additional experiments to address the reviewer's comments.

Referee #3:

Impaired autophagy has been shown to cause mitochondrial dysfunction and heart failure. While altered mitophagy and protein quality control contribute to, it is not known whether other mechanisms involved in impaired autophagic flux could explain mitochondrial dysfunction and heart failure. In this study, the authors show that Atg3 deficiency-induced impaired autophagy led to accelerated NAD⁺ clearance and NAD⁺ deficiency. This contributes to the development of mitochondrial and cardiac dysfunction and resultant heart failure. Overall, this is a comprehensive, well documented study. The authors used several models and approaches to decipher the mechanisms. The mainly evaluated the components of the NAD⁺ pathway on mRNA level, protein levels and some activities. They also used beside Atg3-deficient mice, experiments in H9c2 cells to test the consequences of activation (torin 2) or inhibition (chloroquine) of autophagy on the OCR or NNMT, as well as luciferase assay for NF- κ B. The authors proposed that lower NAD⁺ levels in autophagy inhibition result from the activation of the p62-NF- κ B-NNMT signal transduction.

Major comments

Q1: 1. The authors claim that the absence of conversion of LC3-I to -II indicates impaired autophagic flux. It would be nice to validate whether inhibition of the autophagic flux in wild-type mice (e.g. with leupeptin) produces similar effects on increased NAD⁺ clearance rate than in Atg3-deficient mice.

Response: We thank the referee for this comment. We have treated mice with chloroquine (CQ) to inhibit autophagic flux as shown in our previous study (Harris *et al*, 2022). In addition, CQ treatment also blocks autophagic flux in H9c2 cells as shown in Fig 6E, which leads to increased binding of NF κ b to the NNMT promoter (Fig 7F). We have now measured NAD⁺ levels in CQ-treated mouse hearts and observed decreased NAD⁺ levels (Fig 3E). Therefore, inhibition of autophagic flux in wild-type mice with chloroquine produces similar effects on NAD⁺ concentrations, which we believe is the consequence of increased NAD⁺ clearance rate as we also observe in Atg3-deficient mice.

Harris MP, Zhang QJ, Cochran CT, Ponce J, Alexander S, Kronemberger A, Fuqua JD, Zhang Y, Fattal R, Harper T *et al* (2022) Perinatal versus adult loss of ULK1 and ULK2 distinctly influences cardiac autophagy and function. *Autophagy* 18: 2161-2177
Neelakantan H, Brightwell CR, Graber TG, Maroto R, Wang HL, McHardy SF, Papaconstantinou J, Fry CS, Watowich SJ (2019) Small molecule nicotinamide N-methyltransferase inhibitor activates senescent muscle stem cells and improves regenerative capacity of aged skeletal muscle. *Biochem Pharmacol* 163: 481-492
Neelakantan H, Vance V, Wetzel MD, Wang HL, McHardy SF, Finnerty CC, Hommel JD, Watowich SJ (2018) Selective and membrane-permeable small molecule inhibitors of nicotinamide N-methyltransferase reverse high fat diet-induced obesity in mice. *Biochem Pharmacol* 147: 141-152

Q2: 2. The ratio of LC3-II/LC3-I is not accepted by the Autophagy guidelines. Therefore, it would be better to relate LC3-II on GAPDH only.

Response: We thank this referee for this suggestion. We have recalculated the ratio of LC3-I on GAPDH and LC3-II on GAPDH in related Figures (Fig 1F, Fig 6A, 6E, 6K, and 6M).

Q3: 3. Several data, particularly for the pathway analysis (Figure 4 and 5) are done on variable number of mice (for example Figure 5A: n=7,6, then 8 for WT and n=5,6 and 5 for cAtg3KO mice). I would be more secure in my review to see the same number of mice in all panels.

Response: We thank this referee for this comment. While this is optimal, we are dependent on the availability of mice at specific ages for the multiple experiments that are conducted. We always strived for a minimal number of mice of 5 per group for each experiment, but if additional mice were available, then we included them in the respective experiments. No data

were removed.

Q4: 4. The loading control GAPDH for detection of NNMT is not appropriate, because of very close MW. Therefore, I have doubt that both were done on the same membrane shown in Figures 6 and 7. Full blots would be good in the supplemental file.

Response: The molecular weights of GAPDH and NNMT are 36 and 27 kDa, respectively. These two proteins may be clearly separated on immunoblotting. In addition, our acquisition of immunoblots was completed using Odyssey DLx (Li-COR) composed of two channels, 700 nm and 800 nm lasers. These two channels can scan immunoblots independently and simultaneously. In our immunoblots, the primary antibody against GAPDH was raised in mice, and the NNMT primary antibody was raised in rabbits. These two immunoblots were completed in different channels. In addition, we provide full blots in supplementary files.

Q5: 5. Figure 7F: would be nice to see the CHIP in this condition.

Response: The data in Figure 7F was obtained by CHIP, where we used real-time PCR to amplify the DNA sequences to which there was increased binding. We did not run a gel. Figure 7E, illustrates that NF- κ B binds to the NNMT promoter by CHIP.

Q6: 6. Figure 7M: is the cp62 KO associated with higher autophagy activity?

Response: We thank this referee for this comment. Autophagic flux in cp62 KO mouse hearts has been evaluated in a separate manuscript (revision under review in *Cardiovasc Res*). These data indicate that basal autophagy is no different between WT and cp62 KO mice. Moreover, we also show in Figure 7K of the revised manuscript that overexpression of p62 does not alter LC3II levels. Taken together these data indicate that altering p62 levels alone, are not sufficient to alter autophagy.

Q7: 7. Do you have any evidence of cell death in vitro (Atg3si or chloroquine)?

Response: We thank this referee for this comment. We have conducted new experiments to determine the rate at which cell viability is altered when autophagic flux is impaired by chloroquine or following Atg3 silencing, see (Fig EV9). Note that studies of NAD⁺ levels were conducted either immediately after Atg3 silencing or within 16h of chloroquine treatment. At these time points there was no evidence of cell death. In the presence of CQ or following Atg3 knockdown although can detect increased apoptosis as evidenced by cleaved caspase 3, evidence of reduced cell viability based on the number of live cells, is seen only at 24 hours and beyond.

Q8: 8. Is mitophagy defective in Atg3KO mice?

Response: We thank this referee for this comment. We evaluated mitophagy in WT vs cAtg3 KO mouse hearts. As shown in Figure 2H and I, mitophagy is impaired in cAtg3 KO mouse hearts as evidenced by decreased Parkin expression and reduced LC3-II levels in the mitochondrial fraction.

Minor comments

Q9: 1. EV4: representative graph - correct NANMAT1-3 by NMNAT1-3.

Response: We thank this referee for careful reading. We have corrected these misspellings.

Q10: 2. MW markers should be shown in all blots, including the EV.

Response: We thank this referee for this suggestion. We have added MW markers to all blots.

Q11: 3. In the introduction some references to other paper showing autophagy alteration in human or mouse models of inherited cardiomyopathy would be nice.

Response: We thank this referee for this suggestion. We have cited some references showing autophagy alteration in human or mouse models of inherited cardiomyopathy, in the introduction of the revised manuscript.

Referee #3, cross-comments

Q12: To me the rationale was clear even if ATG3 has not been shown to play a role in humans, a mouse model of Atg3 deficiency to mimic autophagy impairment is good. Still and as I said in my review, I think the evaluation of the autophagic flux is incomplete. And I agree for additional genetic support for a NNMT dependence of autophagy-controlled NAD⁺.

Response: Thank you for this feedback. Given that NNMT KO mice are not immediately available, and a small molecule NNMT inhibitor (NNMTi, 5-Amino-1-methylquinolinium iodide) has been successfully used in some mouse experiments (Neelakantan *et al*, 2019; Neelakantan *et al*, 2018), we administered NNMTi to 30-week-old cAtg3 KO mice and measured cardiac function. As shown in Figure 8E to F, the administration of NNMTi significantly restored NAD levels and rescued cardiac function in cAtg3 KO mouse hearts without altering autophagic defects in these hearts. This experiment supports our premise that induction of NNMT is the basis for reduction of NAD⁺ in cAtg3 KO mouse hearts, which contributes to cardiac dysfunction.

Neelakantan H, Brightwell CR, Graber TG, Maroto R, Wang HL, McHardy SF, Papaconstantinou J, Fry CS, Watowich SJ (2019) Small molecule nicotinamide N-methyltransferase inhibitor activates senescent muscle stem cells and improves regenerative capacity of aged skeletal muscle. *Biochem Pharmacol* 163: 481-492

Neelakantan H, Vance V, Wetzel MD, Wang HL, McHardy SF, Finnerty CC, Hommel JD, Watowich SJ (2018) Selective and membrane-permeable small molecule inhibitors of nicotinamide N-methyltransferase reverse high fat diet-induced obesity in mice. *Biochem Pharmacol* 147: 141-152

Referee #2, cross-comments

I also agree that the as a model of autophagy deficiency Atg3 KO is justified here. Some effects might be slight but considering the data is from animals it still relevant, conceptually it's an interesting story providing novel link between autophagy and NAD metabolism. I think autophagy flux defect is shown sufficiently, this is a KO of an

essential autophagy gene and LC3/p62 blots/quantifications show clear autophagy defect. The cross with NNMT mouse would be nice but depends whether those are available.

Response: Thank you for this feedback. Given that NNMT KO mice are not immediately available, and a small molecule NNMT inhibitor (NNMTi, 5-Amino-1-methylquinolinium iodide) has been successfully used in some mouse experiments (Neelakantan *et al.*, 2019; Neelakantan *et al.*, 2018), we administered NNMTi to 30-week-old cAtg3 KO mice and measured cardiac function. As shown in Figure 8E to F, the administration of NNMTi significantly restored NAD levels and rescued cardiac function in cAtg3 KO mouse hearts without altering autophagic defects in these hearts. This experiment supports our premise that induction of NNMT is the basis for reduction of NAD⁺ in cAtg3 KO mouse hearts, which contributes to cardiac dysfunction.

Dear Dr Abel,

Thank you for submitting your revised manuscript (EMBOJ-2023-113557R) to The EMBO Journal. As mentioned, your amended study was sent back to the referees for their re-evaluation, and we have received comments from two of them, which I enclose below. Please note that while referee #1 was unfortunately not able to re-evaluate the revised work at this time, we have assessed your response to the concerns raised editorially and found them to be addressed satisfactorily. As you will see, the other experts stated that the work has been substantially improved by the revisions and they are now broadly in favour of publication, pending minor revision.

Thus, we are pleased to inform you that your manuscript has been accepted in principle for publication in The EMBO Journal.

Please consider the remaining discussion points by referee #2 regarding nomenclature and literature references carefully and adjust the text where appropriate.

Also, we now need you to take care of a number of minor issues related to formatting and data presentation as detailed below, which should be addressed at re-submission.

Please contact me at any time if you have additional questions related to below points.

As you might have noted on our web page, every paper at the EMBO Journal now includes a 'Synopsis', displayed on the html and freely accessible to all readers. The synopsis includes a 'model' figure as well as 2-5 one-short-sentence bullet points that summarize the article. I would appreciate if you could provide this figure and the bullet points.

Thank you for giving us the chance to consider your manuscript for The EMBO Journal. I look forward to your final revision.

Again, please contact me at any time if you need any help or have further questions.

Best regards,

Daniel Klimmeck

>> Adjust the title of the 'Conflict of Interests' section to 'Disclosure and Competing Interests Statement'.

>> Author Contributions: Please remove the author contributions information from the manuscript text. Note that CRediT has replaced the traditional author contributions section as of now because it offers a systematic machine-readable author contributions format that allows for more effective research assessment. and use the free text boxes beneath each contributing author's name to add specific details on the author's contribution.

More information is available in our guide to authors.

>> Figures files: main figures should be uploaded as individual, high-resolution figure files in TIF, EPS or PDF format.

>> Funding information: please update the information provided in our online system. Currently missing: Fraternal Order of Eagles Diabetes Research Center, Roy J. Carver Trust, Alfred E. Mann Family Foundation, AHA award 20CDA35320042.

>> Appendix file: The supplemental figure files should be merged with their legends into one PDF and renamed "Appendix Figure S1" etc. The appendix will need a ToC with page numbers on the first page. There is the option to make up to 5 EV Figures. The figure files will need to be uploaded as individual, high-resolution figure files and labeled "Figure EV1" etc., and the legends will need to be added to the manuscript, after the main figure legends. The source data files and figure callouts in the

manuscript will need to be adjusted accordingly.

>> Data not shown: please provide respective data or remove the statement on p.10 from the manuscript.

>> Please indicate data re-use in the legends for the Western blot data presented in Figures 3C and EV6D.

>> Please annotate antigen retargeting in the legend for Figure EV11A ('GFP-LC3-I,II' vs 'LC3-I,II').

>> Data availability section: please add a separate data availability section stating: 'No large-scale data amenable to repository deposition were generated in this study.' .

>> Consider additional changes and comments from our production team as indicated below:

- Figure legends:

1. Please note that a separate 'Data Information' section is required in the legends of figures 1-4.
2. Please indicate the statistical test used for data analysis in the legends of figures 1b, c, f, g-i; 2a-o; 3a-b, d-e; 4b-f; 5a-g, i-j; 6a-b, d-n; 7a-d, f-o; 8a-e, g, i; EV2b; EV3b; 4a; EV6a-e; EV8a-f; EV9a-d
3. Please define the annotated p values ****/*/#** in the legend of figure 7a-d, f-o; 8a-c, e, g, i; EV7d as appropriate.
4. Please note that the error bars are not defined in the legend of figures 7a-d, f-o; 8a-c, e, g-i; EV6b; EV7a
5. Please note that scale bar and its definition are missing for figures EV3a; EV9b, d
6. Please indicate what arrows represent in figure EV3a

Referee #2:

This reviewer believes that the authors sufficiently addressed all the criticisms and provided convincing new data which significantly strengthened the manuscript. Overall, the manuscript presents solid body of data with a strong impact for the field.

It is not clear how the multiple files in the EV figures will be presented in the publication, the Editors should be able to advise. The authors should also carefully proofread the manuscript and correct gene/protein names throughout the text and figures (please use HUGO nomenclature as a reference). Mouse and human genes/proteins should be presented in respective formats (sentence case and all capitals, respectively). p62 should be in sentence case (please check new figures), similarly correct abbreviations for p70-S6K etc. Gene names should be italicised. Finally, bidirectional links between autophagy and NAD⁺ have been more extensive than the only reference the authors are citing in the introduction. Using the recent review on the topic DOI: 10.1016/j.tcb.2023.02.004 could be advised.

Referee #3:

The authors addressed all my points, including additional experiments, also suggested by other referees, e.g. using NNMT inhibitor in vivo. This has markedly improved the manuscript. I do not have any other concern.

The authors addressed the remaining editorial issues.

Dear Dr Abel,

Thank you for submitting the revised version of your manuscript. I have now evaluated your amended manuscript and concluded that the remaining minor concerns have been sufficiently addressed.

Thus, I am pleased to inform you that your manuscript has been accepted for publication in the EMBO Journal.

Please note that it is The EMBO Journal policy for the transcript of the editorial process (containing referee reports and your response letter) to be published as an online supplement to each paper.

If you do NOT want the transparent process file published, you will need to inform the Editorial Office via email immediately. More information is available here: https://www.embopress.org/transparent-process#Review_Process

On a different note, I would like to alert you that EMBO Press offers a format for a video-synopsis of work published with us, which essentially is a short, author-generated film explaining the core findings in hand drawings, and, as we believe, can be very useful to increase visibility of the work. This has proven to offer a nice opportunity for exposure i.p. for the first author(s) of the study. Please see the following link for representative examples and their integration into the article web page:

<https://www.embopress.org/doi/full/10.15252/emj.2019103932>

If you have any questions, please do not hesitate to call or email the Editorial Office.

Best regards,

Daniel Klimmeck

Daniel Klimmeck, PhD
Senior Editor
The EMBO Journal
EMBO
Postfach 1022-40
Meyerhofstrasse 1
D-69117 Heidelberg
contact@embojournal.org
Submit at: <http://emboj.msubmit.net>